# AN INTUITIVE MULTI-FREQUENCY FEATURE REPRESENTATION FOR SO(3)-EQUIVARIANT NETWORKS

**Dongwon Son, Jaehyung Kim, Sanghyeon Son, Beomjoon Kim**
Department of AI, KAIST
{dongwon.son,kimjaehyung,ssh98son,beomjoon.kim}@kaist.ac.kr

## ABSTRACT

The usage of 3D vision algorithms, such as shape reconstruction, remains limited because they require inputs to be at a fixed canonical rotation. Recently, a simple equivariant network, Vector Neuron (VN) (Deng et al., 2021) has been proposed that can be easily used with the state-of-the-art 3D neural network (NN) architectures. However, its performance is limited because it is designed to use only three-dimensional features, which is insufficient to capture the details present in 3D data. In this paper, we introduce an equivariant feature representation for mapping a 3D point to a high-dimensional feature space. Our feature can discern multiple frequencies present in 3D data, which, as shown by Tancik et al. (2020), is the key to designing an expressive feature for 3D vision tasks. Our representation can be used as an input to VNs, and the results demonstrate that with our feature representation, VN captures more details, overcoming the limitation raised in its original paper.

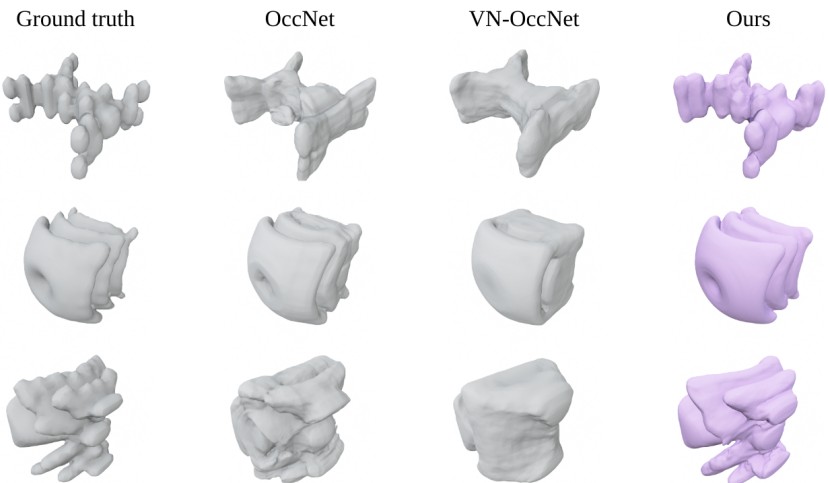

Figure 1: EGAD (Morrison et al., 2020) meshes constructed from the embeddings given by different models based on OccNet (Mescheder et al., 2019) at canonical poses. As already noted in their original paper, VN-OccNet (3rd column), the VN version of OccNet, fails to capture the details present in the ground-truth shapes and does worse than OccNet (2nd column). Using our feature representation, VN-OccNet qualitatively performs better than OccNet (4th column). Note that each of these shapes consists of multiple frequencies – in some parts of the object, the shape changes abruptly, while in some parts, it changes very smoothly.

## 1 INTRODUCTION

SO(3) equivariant neural networks (NN) change the output accordingly when the point cloud input is rotated without additional training. For instance, given a point cloud rotated by, say, 90 degrees,

an SO(3) equivariant encoder can output an embedding rotated by 90 degrees without ever being trained on the rotated input. Pioneering works such as TFN and SE(3) transformers (Thomas et al., 2018; Fuchs et al., 2020) use tools from quantum mechanics to design an equivariant network, but they are difficult to understand and require a specific architecture that is incompatible with recent architectural advancements in 3D vision.

More recently, Vector neuron (VN) (Deng et al., 2021) has been proposed as a more accessible alternative. By extending a neuron, which typically outputs a scalar, to output a three-dimensional vector and modifying typical operations in NNs (e.g. ReLU activation function, max pooling, etc.), VN guarantees equivariance. Further, because it uses the existing operations in NNs, it can easily be implemented within well-established point processing networks like PointNet (Li et al., 2018) and DGCNN (Wang et al., 2019). However, their key limitation is the low dimensionality of the features: unlike TFN or SE(3)-transformers which use high-dimensional features, a feature in vector neurons is confined to a 3D space. This limits the expressivity of the features and cannot capture the details present in 3D data, as depicted in Figure 3.

In this work, we propose an equivariant feature representation for mapping a 3D point to a high-dimensional space. The key challenge here is designing an algorithm that computes expressive yet rotation equivariant features from 3D point clouds. As shown by Mildenhall et al. (2021) and Tancik et al. (2020), for 3D data, the effectiveness of features heavily depends on its ability to represent different frequencies present in the input, as 3D shapes typically are multi-frequency signals (e.g. Figure 1). So, one way is to use a Fourier basis that consists of sinusoids of various frequencies as done in NeRF (Mildenhall et al., 2021). However, this does not guarantee equivariance.

Based on the observation that a rotation matrix can be written as sinusoids whose frequencies are determined by its eigenvalues (Fulton & Harris, 2013), we instead propose to construct a mapping $D : SO(3) \to SO(n)$ and use $D$ to define our feature representation in a way that is provably equivariant. At a high-level, our idea is to describe the orientation of a given point $\vec{u} \in \mathbb{R}^3$ from a basis axis, say $\hat{z}$, denoted $R^{\hat{z}}(\vec{u})$, and then use $D$ to map $R^{\hat{z}}$ to a higher dimensional feature space such that it defines the same amount of rotation from a basis axis in $\mathbb{R}^n$. Then, we simply apply $D(R^{\hat{z}}(\vec{u}))$ to the chosen basis axis in $\mathbb{R}^n$ to get our feature representation of $\vec{u}$. This guarantees equivariance because when we rotate $\vec{u}$, the corresponding feature would also rotate by the same amount. Figure 2 demonstrates this intuition.

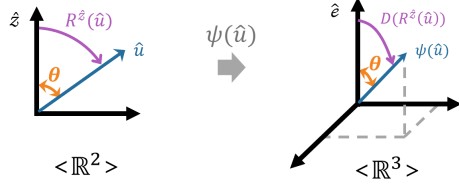

Figure 2: Intuition of our equivariant feature representation, $\psi$, that maps a point in 2D to 3D (i.e. $n = 3$) for an illustrative purpose. (Left) The basis axis in 2D is $\hat{z} = [0, 1]$, and $\hat{u} = R^{\hat{z}}(\hat{u})\hat{z}$, with $\theta$ as its amount of rotation from $\hat{z}$. Our $D$ is constructed so that it defines the same amount of rotation, $\theta$, but from a basis in the 3D space, which in this case is chosen to be $\hat{e} = [0, 0, 1]$. The feature representation of $\hat{u}$ is given by $\psi(\hat{u}) = D(R^{\hat{z}}(\hat{u}))\hat{e}$. As you can see, when $\theta$ changes, it rotates both $\hat{u}$ and $\psi(\hat{u})$ by the same amount. Note that the description of magnitude is neglected for brevity.

We propose a set of mathematical conditions for constructing $D$ that achieves this, and prove that our feature representation is equivariant. Further, we show this rotation-matrix-based feature representation can be written as sinusoids, much like in Fourier bases, whose frequency is determined by the maximum eigenvalues of $D(R)$. Based on this result, we propose an algorithm that can construct $\psi$ that consists of sinusoids of frequency $\lfloor \frac{n-1}{2} \rfloor$.

We call our representation **F**requency-based **E**quivariant feature **R**epresentation (FER). We use it as an input to VN instead of 3D points and integrate VN into standard point processing architectures, PointNet and DGCNN, and show FER-VN-PointNet and FER-VN-DGCNN achieve state-of-the-art performance among equivariant networks in various tasks, including shape classification, part segmentation, normal estimation, point completion, and shape compression. Notably, unlike the standard VN which performs worse than the non-equivariant counterpart in the point completion and compression tasks at a canonical pose, we show that FER-VN outperforms both of them by capturing the high-frequencies present in the 3D shape as illustrated in Figure 3.

## 2 RELATED WORKS

### 2.1 $SO(3)$-EQUIVARIANT NETWORKS

For 3D inputs, several equivariant methods have been proposed (Weiler et al., 2018; Thomas et al., 2018; Fuchs et al., 2020; Brandstetter et al., 2021; Anderson et al., 2019). These models use an equivariant feature representation called spherical harmonics to map 3D points to an $n$ dimensional feature space where rotations are described using Wigner-D matrices. While this representation is equivariant, this Wigner-D matrix is from quantum mechanics originally used to describe quantum states (Sakurai & Commins, 1995), and it is difficult to understand why they work, and what the intuitions behind the hyperparameters are. For instance, what does the number of dimensions of representations imply? Does it relate to the intuition that we need to discern multiple frequencies? For readers without a background in quantum mechanics, these questions are hard to answer. Furthermore, these methods are confined to using specific architectures, making it difficult to leverage state-of-the-art 3D vision architectures. One of our goals in this work is to offer a more accessible and intuitive alternative to derive a high-dimensional frequency-based equivariant feature representation.

Recent advancements focus on creating more adaptable $SO(3)$-equivariant neural networks, especially for point cloud processing (Deng et al., 2021; Puny et al., 2021; Kaba et al., 2023). Frame averaging method (Puny et al., 2021) proposes to achieve equivariance by averaging over a group through symmetrization, but its use of Principal Component Analysis (PCA) makes it sensitive to noisy and missing points, a phenomenon prevalent in 3D sensors. Kaba et al. (2023) introduces a method for an equivariant structure by training an extra network that maps rotated inputs to a canonical pose. However, a small error in this network could have a large negative impact on the overall performance. Among these, Vector Neurons Deng et al. (2021) stands out because it achieves equivariance by simply modifying the existing operations in NNs. Its key limitation however is the confinement to the 3D feature space. Our work offers a high dimensional frequency-based representation that can be used with VNs.

### 2.2 FREQUENCY-BASED FEATURE REPRESENTATION

Frequency-based sinusoidal feature representation has been widely adopted in 3D vision. Perhaps the most well-known usage is in NeRF (Mildenhall et al., 2021), where it maps a viewing position and angle into a high-dimensional, multi-frequency sinusoidal feature representation. Similarly, Tancik et al. (2020) shows that using sinusoidal feature representation achieves significantly better performance in shape compression task than using 3D coordinate as an input to the network. The reason this works better than the coordinate-based representation is that, for a complex shape, some parts are very non-smooth and change rapidly with respect to the changes in the position on the shape, while some parts are smooth (e.g. shapes in Figure 1). Using sinusoidal feature representation, such as Fourier basis, at different frequencies helps because they can discern both smooth and non-smooth changes. However, the conventional Fourier basis is not SO(3) equivariant. Our feature representation guarantees equivariance and captures multiple frequencies

## 3 FREQUENCY-BASED EQUIVARIANT FEATURE REPRESENTATION

Our feature representation $\psi : \mathbb{R}^3 \to \mathbb{R}^n$ is defined as

$$\psi(\vec{u}) = ||\vec{u}||D(R^z(\hat{u}))\hat{e} , \tag{1}$$

where $\vec{u}$ is the input point, $\hat{e}$ is a basis in the $n$ dimensional space, $R^z(\hat{u})$ is a rotation matrix defining the orientation measured from z-axis (i.e. $[0,0,1]$) to $\hat{u}$. We will show that by construction, $\psi$ is provably equivariant, and that it consists of sinusoids whose maximum frequency is $\lfloor \frac{n-1}{2} \rfloor / 2\pi$. Let us first describe the conditions and construction of $D$ that make this possible. All proofs for theorems and propositions in this section are in the appendix.

First and foremost, we would like $D(R)$ to be an element of $SO(n)$. Second, we would like to map every element of $R \in SO(3)$ to a unique element in $SO(n)$ so that changes in $SO(3)$ are uniquely

captured in $SO(n)$. This gives us the following set of conditions for $D$.

$$
\begin{aligned}
&\forall R_1, R_2 \in SO(3), \text{if } R_1 \neq R_2, \text{then } D(R_1) \neq D(R_2), \\
&\forall R \in SO(3), D \text{ maps } R \text{ to a single matrix } D(R) \in \mathbb{R}^{n \times n}, \\
&\forall R \in SO(3), D(R)D(R)^T = D(R)^T D(R) = I, \\
&\forall R \in SO(3), \det(D(R)) = 1.
\end{aligned}
\tag{2}
$$

The first two conditions ensure that $D$ uniquely maps for all $R \in SO(3)$. The last two conditions ensure that $D(R)$ belongs to $SO(n)$ (Stillwell, 2008).

To construct $D$ that satisfies these conditions, we use the exponential form of a rotation matrix. In $\mathbb{R}^3$, given $R \in SO(3)$ expressed in axis-angle representation with $\hat{w} \in \mathbb{R}^3$ as its rotation axis and angle $\theta \in \mathbb{R}$, we have $R = \exp(\theta \hat{w} \cdot \vec{F})$ where $\hat{w} \cdot \vec{F} = \sum_{i=1}^{3} \hat{w}_i F_i$, $\hat{w}_i$ as $i^{th}$ element of $\hat{w}$, and $\vec{F} = [F_1, F_2, F_3]$,

$$
F_1 = \begin{bmatrix} 0 & 0 & 0 \\ 0 & 0 & -1 \\ 0 & 1 & 0 \end{bmatrix}, F_2 = \begin{bmatrix} 0 & 0 & 1 \\ 0 & 0 & 0 \\ -1 & 0 & 0 \end{bmatrix}, F_3 = \begin{bmatrix} 0 & -1 & 0 \\ 1 & 0 & 0 \\ 0 & 0 & 0 \end{bmatrix}
$$

Intuitively, $F_1, F_2, F_3$ describe the rotation about the $x, y, z$ axes respectively (Fulton & Harris, 2013). For instance, when $\hat{w} = [1, 0, 0]$, then $R$ is determined solely by $F_1$.

We generalize this representation to $n$ dimensional space, where we use $\vec{J} = [J_1, J_2, J_3]$, $J_i \in \mathbb{R}^{n \times n}$, so that

$$
D(R) = \exp(\theta \hat{w} \cdot \vec{J})
\tag{3}
$$

where $\theta$ and $\hat{w}$ denote the angle and axis of rotation of $R \in SO(3)$, respectively. Intuitively, just like $F_1, F_2$ and $F_3$ represent axes of rotation about $x, y$ and $z$ in $\mathbb{R}^3$, $J_1, J_2$ and $J_3$ represent the angles about axes $\psi(\hat{x}), \psi(\hat{y})$, and $\psi(\hat{z})$. The key here is that they describe the same amount of rotation angle even when they are in different spaces. For instance, if we set $\theta = [0, 0, \theta_z]$ and $\hat{w} = [0, 0, 1]$, then the only contributing term for $D(R)$ is $\theta_z J_3$. Based on this observation, we will construct $\vec{J}$ instead of constructing $D$ directly. We have the following helpful theorem.

**Theorem 1.** *If $J_i \in \mathbb{R}^{n \times n} \ \forall i \in \{1, 2, 3\}$ satisfies $-J_i = J_i^T$, $[J_1, J_2] = J_3$, $[J_2, J_3] = J_1$, $[J_3, J_1] = J_2$ where $[A, B] = AB - BA$, and $\exp(2m\pi J_i) = I_{n \times n}, \forall m \in \mathbb{Z}$, where $\mathbb{Z}$ is the space of integers, then $D(R) = \exp(\theta \hat{w} \cdot \vec{J}) \in SO(n)$ and satisfies the conditions in Equation 2.*

Given these conditions, we can design an algorithm for constructing $\vec{J}$. But before doing so, let us show the relationship between the eigenvalues of $D(R)$ and the frequency of $\psi$. This will provide a way to design $\vec{J}$ so that $\psi$ has the maximum possible frequency for a given $n$. The following propositions and theorem will help us do that.

**Proposition 1.** *Suppose $J_1, J_2, J_3$ satisfy the conditions in Theorem 1. Then, $J_1, J_2$, and $J_3$ have the same eigenvalues and multiplicities. In particular, the eigenvalues are $\Lambda = \{-k\mathrm{i}, -(k-1)\mathrm{i}, \ldots, -\mathrm{i}, 0, \mathrm{i}, \ldots, k\mathrm{i}\}$ for some non-negative integer $k$. Further, if $\lambda$ is an eigenvalue of $J_i$ with multiplicity $m$, then $-\lambda$ is also an eigenvalue of $J_i$ with the same multiplicity $m$.*

**Proposition 2.** *Suppose $J_1, J_2, J_3$ satisfy the conditions in Theorem 1. Then, $\theta \hat{w} \cdot \vec{J}$ have the same eigenvalues as $J_i$, $\Lambda = \{-k\theta\mathrm{i}, -(k-1)\theta\mathrm{i}, \ldots, -\theta\mathrm{i}, 0, \theta\mathrm{i}, \ldots, k\theta\mathrm{i}\}$*

**Theorem 2.** *Suppose $D$ satisfies the conditions in Equation 2, and $J_1, J_2, J_3$ satisfy the conditions in Theorem 1. $\forall R \in SO(3)$ whose angle of rotation is $\theta$ and rotation axis is $\hat{w}$, we have*

$$
D(R) = \exp(\theta \hat{w} \cdot \vec{J}) = \sum_{\lambda \in \Lambda} \vec{b}_\lambda \vec{b}_\lambda^T (\sin(\lambda\theta) + \mathrm{i} \cos(\lambda\theta))
$$

*where $\vec{b}_\lambda$ is the eigenvector of $\theta \hat{w} \cdot \vec{J}$ that corresponds to eigenvalue $\lambda$.*

Note the correlation between the frequency of $D(R)$ and the magnitude of $\lambda$. If we have a large $\lambda$, then $D(R)$ consists of sinusoids with large magnitude, and vice-versa. The following proposition gives us insight as to why high-dimensional features are necessary.

**Proposition 3.** *The maximum value of $k$ in Proposition 2 is $\lfloor \frac{n-1}{2} \rfloor$.*

Therefore, for all $R \in SO(3)$, the maximum frequency $D(R)$ can have is $\lfloor \frac{n-1}{2} \rfloor / 2\pi$. This shows that if our feature space is say, dimension 3, then the maximum frequency it can attain is just $1/2\pi$.

Finally, we have the following equivariance theorem for FER.

**Theorem 3.** *Consider the mapping $\psi(\vec{u}) = ||\vec{u}|| D(R^z(\hat{u}))\hat{e}$. $\psi$ is rotational equivariant if $\hat{e}$ is the eigenvector corresponding to the zero eigenvalue of $J_3$, and $D$ satisfies all the conditions in equation 2.*

The reason that we have the condition for $\hat{e}$ is that we are using the angle from the z-axis in 3D space to $\hat{u}$, $R^z(\hat{u})$, to express its orientation. Note that we could have chosen instead to use $R^x(\hat{u})$ or $R^y(\hat{u})$ in $\psi$. In such cases, we would have to choose $\hat{e}$ as an eigenvector with the zero eigenvalues of $J_1$ or $J_2$ respectively.

We now have all the ingredients for constructing $D$. We will ensure to construct $D$ to satisfy all the conditions in Theorem 1, and $\lfloor \frac{n-1}{2} \rfloor$i is included in $\vec{J}$ so that it can capture the maximum frequency with the given dimension $n$. Algorithm 1 gives a pseudocode for doing this.

---

**Algorithm 1** CONSTRUCT $J_1, J_2, J_3$

**Require:** $n$
1: $J_3 \leftarrow$ SAMPLE$J_3(n)$
2: $\mathcal{J}_1, \mathcal{J}_2 \leftarrow$ CREATESEARCHSPACE$(J_3)$
3: $J_1, J_2 \leftarrow \underset{J_1 \in \mathcal{J}_1, J_2 \in \mathcal{J}_2}{\operatorname{argmin}} ||[J_1, J_2] - J_3||_F^2$
4: **return** $[J_1, J_2, J_3]$

---

**Algorithm 2** SAMPLE$J_3$

**Require:** $n$
1: $A \leftarrow$ randomly fill $\mathbb{R}^{n \times n}$ matrix
2: $J_3 \leftarrow A - A^T$
3: $U \leftarrow$ GETEIGENVECTORS$(J_3)$
4: $\Lambda =$ DIAGONAL $\left( \left[ -\lfloor \frac{n-1}{2} \rfloor \text{i}, \ldots, \lfloor \frac{n-1}{2} \rfloor \text{i} \right] \right)$
5: $J_3 \leftarrow U\Lambda U^*$
6: **return** $J_3$

---

Given $n$, the algorithm begins by constructing $J_3$ by calling SAMPLE$J_3$. Algorithm 2 begins with a construction of a random skew-symmetric matrix (L1-2), and set its eigenvalues to $\Lambda = \left[ -\lfloor \frac{n-1}{2} \rfloor \text{i}, \ldots, \lfloor \frac{n-1}{2} \rfloor \text{i} \right]$ while keeping the eigenvectors unchanged (L3-5). Given that the returned $J_3$ has eigenvalues $\Lambda$ and is skew-symmetric, we have that $\exp(2k\pi J_3) = I_{n \times n}$ by Proposition 4 in the appendix.

We then return to Algorithm 1 to construct $J_1$ and $J_2$, such that they satisfy conditions in Theorem 1. We now want to find $J_1$ and $J_2$ such that they satisfy

$$[J_3, J_1] = J_2, \; [J_2, J_3] = J_1, \; J_1^T = -J_1 \; [J_1, J_2] = J_3, \tag{4}$$

Note that if we satisfy these conditions, $J_2 + J_2^T = 0$ would be automatically satisfied. To find $J_1$ and $J_2$ that satisfy them, we first define the space of $J_1$ and $J_2$ that satisfies the first three conditions, by realizing that these equations define an under-determined linear system of equations whose unknowns are the elements of $J_1$ and $J_2$ (for details, see Appendix F) (L2). Then, to satisfy the last condition, we solve the non-linear optimization problem using the Cross-Entropy Method (CEM) (De Boer et al., 2005) (L3).

## 4 EXPERIMENT

We conduct our experiments on both SO(3)-invariant task and SO(3)-equivariant task. We use three tasks adopted from Deng et al. (2021): point cloud classification (invariant), segmentation (invariant), and point cloud completion (equivariant in the encoder, invariant in the decoder). Further, we evaluate them on three more SO(3) equivariant tasks: shape compression, the normal estimation, and point cloud registration, adopted from Mescheder et al. (2019), Puny et al. (2021), and Zhu et al. (2022). In all subsections, we call our approach FER-VN.

## 4.1 POINT CLOUD COMPLETION

The objective here is to reconstruct the shape, expressed with a neural implicit function, from a partial and noisy input point cloud. The architecture and training details can be found in Appendix E.2.2.

| Method | I/I | I/SO(3) | SO(3)/SO(3) |
|---|---|---|---|
| OccNet (Mescheder et al. (2019)) | 71.4 | 30.9 | 58.2 |
| VN-OccNet (Deng et al. (2021)) | 69.3 | 69.3 | 68.8 |
| FER-VN-OccNet (Ours) | **71.9** | **71.9** | **71.9** |

Table 1: Volumetric mIoU on ShapeNet reconstruction with neural implicit. These results are average category mean IoU over 9 classes. $I/I$ indicates train and test with canonical poses, $I/SO(3)$ indicates train with canonical poses and test with different $SO(3)$ rotations, and $SO(3)/SO(3)$ indicates train and test with different $SO(3)$ rotations. Bold is the best performance.

**Dataset:** We use ShapeNet consisting of 13 major classes, following the categorization in Deng et al. (2021). The input point cloud $P$ is comprised of 300 points, sampled from each model's surface and perturbed with noise $\epsilon \sim N(0, 0.005)$. Query points for ShapeNet are uniformly sampled within the Axis-Aligned Bounding Box (AABB).

**Results:** Table 1 shows the volumetric mean IoU for reconstruction. As already noted in the original paper (Deng et al., 2021), VN-OccNet performs worse than OccNet at canonical poses. FER-VN-OccNet on the other hand, outperforms both approaches by utilizing frequency-based features. This is evident in Figure 3, where, unlike Vector Neurons, our method captures details present in cars (wheels, side mirrors) and chair's legs. We further demonstrate that FER enhances model robustness across various sample sizes, detailed in Appendix G.1.

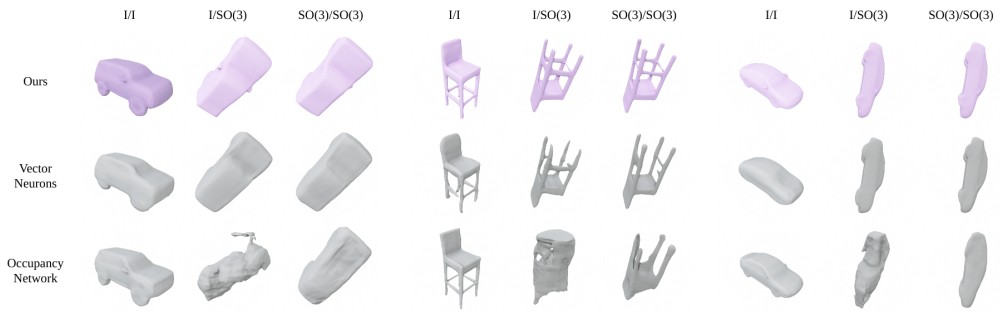

Figure 3: Reconstructions of meshes from point cloud inputs across three models: the original OccNet Mescheder et al. (2019) (bottom), VN-OccNet Deng et al. (2021) (middle), and our proposed model (top).

## 4.2 SHAPE COMPRESSION

Here, the task is to compress a 3D shape into an embedding, and then reconstruct the same shape when just given an embedding. The purpose of this experiment is to investigate how well different models capture details, and in particular, whether our network can learn to capture high-frequency details and compress them to an embedding.

**Dataset:** We use EGAD dataset (Morrison et al., 2020) comprised of 2281 shapes. EGAD dataset provides the categorization of the shapes by shape complexity level from 0 to 25, measured by the distribution of curvature across vertices. We train our model on the given dataset, and save the embeddings. We then reproduce original shapes based on just the embeddings. We use OccNet as the basis model. The training point cloud $P$ is sampled following the strategy in section 4.1. The query points are sampled both from the surface and AABB at a 9:1 ratio, with surface points perturbed by the noise $\delta \sim N(0, 0.025)$. For all models, we train with various different rotations.

**Results:** Figure 4 shows the results. As we can see, as the complexity of the shape increases, the performances of VN drop significantly, while that of ours drops at a slower rate, which again demonstrates the effectiveness of our frequency-based representation. Figure 1 shows the qualitative result at the canonical pose. Our model is able to reconstruct the high-frequency details of the shapes, while VN-OccNet and standard OccNet that use coordinate-based inputs smooth out

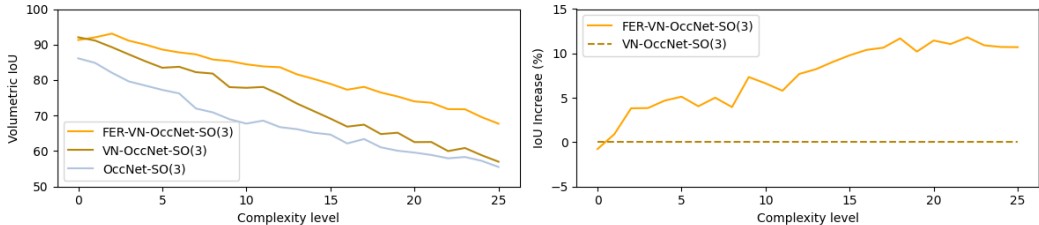

Figure 4: The left graph shows the volumetric IoU of OccNet, VN-OccNet, and FER-VN-OccNet across the complexity level in the EGAD training set. We apply rotational augmentation during both training and test time. The right graph shows FER-VN-OccNet's IoU improvement over VN-OccNet.

or miss the detail, again showing the effectiveness of the frequency-based representation. Dimensional analysis reveals that FER enhances detail accuracy with reduced inference impact, detailed in Appendices G.3, G.2, and G.6. Additionally, we demonstrate improved IoU of our model with consistent compression ratios in Appendix G.4 and report advancements over a recent VN-based model in Appendix G.7.

### 4.3 NORMAL ESTIMATION

The goal of this equivariant task is predicting the normal direction of a point cloud. The ModelNet40 (Wu et al., 2015) and ShapeNet (Chang et al., 2015) datasets are used. For each object, we sampled 512 random surface points. As shown in Table 2, our method outperforms both Vector Neurons and a preceding rotation-equivariant method by Puny et al. (2021).

| dataset | ShapeNet | | ModelNet40 | |
|---|---|---|---|---|
| train/test data | I/SO(3) | SO(3)/SO(3) | I/SO(3) | SO(3)/SO(3) |
| PointNet (Qi et al. (2017a)) | 0.214 | 0.294 | 0.141 | 0.112 |
| VN-PointNet (Deng et al. (2021)) | 0.216 | 0.219 | 0.150 | 0.148 |
| FA-PointNet (Puny et al. (2021)) | 0.208 | 0.215 | 0.152 | 0.156 |
| FER-VN-PointNet (Ours) | 0.196 | 0.198 | 0.125 | 0.126 |
| DGCNN (Wang et al. (2019)) | 0.212 | 0.170 | 0.201 | 0.109 |
| VN-DGCNN (Deng et al. (2021)) | 0.152 | 0.152 | 0.092 | 0.088 |
| FA-DGCNN (Puny et al. (2021)) | 0.150 | 0.153 | 0.083 | 0.082 |
| FER-VN-DGCNN (Ours) | **0.143** | **0.142** | **0.078** | **0.079** |

Table 2: Test normal estimation results on the ShapeNet and ModelNet40 dataset. Numbers indicate the evaluation metric adopted from Puny et al. (2021) $1 - (\mathbf{n}^T \hat{\mathbf{n}})^2$ where is prediction $\hat{\mathbf{n}}$ and $\mathbf{n}$ is ground-truth. Bold is the best performance.

### 4.4 POINT CLOUD REGISTRATION

The objective of point cloud registration is to align two sets of points $P_1$ and $P_2$ that come from the same shape. Registration algorithms such as ICP typically try to find the correspondences between the points in $P_1$ and $P_2$ but they are prone to noise. Zhu et al. (2022) proposes a different approach, which solves for the latent codes $Z_1, Z_2 \in \mathbb{R}^{C \times 3}$ using VNs. Following this, we use the encoders of various models from the shape completion task (section 4.1), find the orientation in the feature space, and from this determine $R$. The details can be found in Appendix E.2.4.

**Dataset:** We use ShapeNet Chang et al. (2015). We generate two point clouds sampled from the same shape, with one being randomly rotated from its canonical pose. Unlike the studies by Zhu et al. (2022) and Yuan et al. (2020), which employ the ModelNet40 dataset and generate denser point clouds of 1,000 or 500 points, we opt for a sparser point cloud of 300 points using ShapeNet. Our experiments feature three test configurations: 1) $P_2$ is created by copying $P_1$ (Copy), 2) $P_2$ consists of different points from the same shape from $P_1$ (Distinct sample); 3) $P_2$ consists of a different set of points and density from $P_1$ where the number of points for $P_2$ varies between 50 and 300 points(Varying density). We evaluate our methodology against three baseline approaches: VN-EquivReg (Zhu et al., 2022), DeepGMR (Yuan et al., 2020) following the setting of (Zhu et al., 2022), and Iterative Closest Point (ICP)(Besl & McKay, 1992). Due to the symmetrical characteristics of ShapeNet objects and the absence of point correspondence information, we use Chamfer Distance (CD) as our evaluation metric and skip the training with pose error in Zhu et al. (2022).

**Results:** Our hypothesis posits that our feature's ability to capture input details results in consistent feature encoding and more robust point cloud registration. Table 3 validates this hypothesis. While all learning-based methods are accurate in recovering the original rotation when a rotated copy is provided, our in the other two settings where we use different point samples and have different density in $P_2$ from $P_1$. Figure 5 shows the qualitative results. VN-EquivReg struggles to distinguish

Figure 5: Qualitative results of point cloud registration. Red is $P_1$ and blue is $P_2$ in the "Distinct sample" setting.

| Rotated point cloud | Copy | Distinct sample | Varying density |
|---|---|---|---|
| ICP | 0.01536 | 0.01609 | 0.02059 |
| DeepGMR (Yuan et al. (2020)) | **0.00000** | 0.00769 | 0.01574 |
| VN-EquivReg (Zhu et al. (2022)) | **0.00000** | 0.00560 | 0.01077 |
| FER-VN-EquivReg (Ours) | **0.00000** | **0.00347** | **0.00714** |
| No rotation | 0.00000 | 0.00310 | 0.00611 |

Table 3: Registration results on the ShapeNet dataset. The metric is Chamfer Distance. Bold is the best performance.

|  | Methods | $z/z$ | $z/SO(3)$ | $SO(3)/SO(3)$ |
|---|---|---|---|---|
|  | Point / mesh inputs |  |  |  |
| Neither | PointNet (Qi et al. (2017a)) | 85.9 | 19.6 | 74.7 |
|  | PointNet++ (Qi et al. (2017b)) | 91.8 | 28.4 | 85.0 |
|  | PointCNN (Li et al. (2018)) | 92.5 | 41.2 | 84.5 |
|  | DGCNN (Wang et al. (2019)) | 90.3 | 33.8 | 88.6 |
|  | ShellNet (Zhang et al. (2019b)) | **93.1** | 19.9 | 87.8 |
| Rotation-equivariant | VN-PointNet (Deng et al. (2021)) | 77.5 | 77.5 | 77.2 |
|  | VN-DGCNN (Deng et al. (2021)) | 89.5 | 89.5 | 90.2 |
|  | FER-VN-PointNet (Ours) | 88.2 | 87.8 | 88.8 |
|  | FER-VN-DGCNN (Ours) | 90.5 | 90.5 | 90.5 |
|  | SVNet-DGCNN (Su et al. (2022)) | 90.3 | 90.3 | 90.0 |
|  | TFN (Thomas et al. (2018)) | 88.5 | 85.3 | 87.6 |
|  | Spherical-CNN (Esteves et al. (2018)) | 88.9 | 76.7 | 86.9 |
|  | $a^3$S-CNN (Liu et al. (2018)) | 89.6 | 87.9 | 88.7 |
| Rotation-invariant | SFCNN (Rao et al. (2019)) | 91.4 | 84.8 | 90.1 |
|  | RI-Conv (Zhang et al. (2019a)) | 86.5 | 86.4 | 86.4 |
|  | SPHNet (Poulenard et al. (2019)) | 87.7 | 86.6 | 87.6 |
|  | ClusterNet (Chen et al. (2019)) | 87.1 | 87.1 | 87.1 |
|  | GC-Conv (Zhang et al. (2020)) | 89.0 | 89.1 | 89.2 |
|  | RI-Framework (Li et al. (2021)) | 89.4 | 89.4 | 89.3 |
|  | PaRINet (Chen & Cong (2022)) | 91.4 | **91.4** | **91.4** |

Table 4: Test classification accuracy on the ModelNet40 dataset. Bold is the best performance and underlined is the next best performance.

|  | Methods | $z/SO(3)$ | $SO(3)/SO(3)$ |
|---|---|---|---|
|  | Point / mesh inputs |  |  |
| Neither | PointNet (Qi et al. (2017a)) | 38.0 | 62.3 |
|  | PointNet++ (Qi et al. (2017b)) | 48.3 | 76.7 |
|  | PointCNN (Li et al. (2018)) | 34.7 | 71.4 |
|  | DGCNN (Wang et al. (2019)) | 49.3 | 78.6 |
|  | ShellNet (Zhang et al. (2019b)) | 47.2 | 77.1 |
| Rotation-equivariant | VN-PointNet (Deng et al. (2021)) | 72.4 | 72.8 |
|  | VN-DGCNN (Deng et al. (2021)) | 81.4 | 81.4 |
|  | FER-VN-PointNet (Ours) | 82.7 | 82.1 |
|  | FER-VN-DGCNN (Ours) | 83.4 | 83.5 |
|  | SVNet-DGCNN (Su et al. (2022)) | 81.4 | 81.4 |
|  | TFN (Thomas et al. (2018)) | 76.8 | 76.2 |
| Rotation-invariant | RI-Conv (Zhang et al. (2019a)) | 75.3 | 75.3 |
|  | GC-Conv (Zhang et al. (2020)) | 77.2 | 77.3 |
|  | RI-Framework (Li et al. (2021)) | 79.2 | 79.4 |
|  | PaRINet (Chen & Cong (2022)) | **83.8** | **83.8** |
|  | TetraSphere (Melnyk et al. (2022)) | 82.3 | 82.1 |

Table 5: Test part segmentation results on the ShapeNet dataset. These results are average category mean IoU over 16 classes. Bold is the best performance and underlined is the next best performance.

between similar features, like an airplane's tail and head, or the shade and base of a lamp. In contrast, our method successfully captures finer details, such as the airplane's tailplane and the lamp's cone-shaped shade. We additionally illustrate that FER improves model robustness against diverse initial perturbations, as described in Appendix G.5.

## 4.5 POINT CLOUD CLASSIFICATION AND SEGMENTATION

**Dataset:** For both tasks, we adopted the experimental data setup from Deng et al. (2021). The ModelNet40 dataset (Wu et al., 2015), used for object classification, comprises 40 classes with 12,311 CAD models—9843 are designated for training and the remainder for testing. For the object part segmentation task, the ShapeNet dataset (Chang et al., 2015) was used, containing over 30,000 models spanning 16 shape classes.

**Results:** The results of the object Table 4 contains the results of the object classification experiment, showing the classification accuracy for three different augmentation strategies. Our methodology was benchmarked against multiple rotation-equivariant and -invariant methods. The methods are grouped by whether it is equivariant, invariant, or neither. In the case of rotation-equivariant methods, they originally maintain the orientation on the feature space but achieve invariance via an additional invariant layer. Even though our method is primarily rotation-equivariant, ours is only after the PaRINet (Chen & Cong, 2022) which is specifically designed for invariance, especially in $z/SO(3)$ and $SO(3)/SO(3)$ setups.

## 5 CONCLUSION

In this work, we propose FER, a frequency-based equivariant feature representation tailored for 3D data. Our approach relies on the fact that rotation matrices can be written as sinusods, whose maximum frequency is determined by the dimensionality of the rotation matrix. This was made possible by defining a mapping function $D$ that maps 3D rotation to $SO(n)$ space, such that the basis axes in 3D are preserved in the $n$ dimensional space. When used with VN and state-of-the-art

point processing networks such as PointNet and DGCNN, our experimental results demonstrate that it captures details that previous methods fail to capture in various different 3D vision tasks.

## 6 REPRODUCIBILITY STATEMENT

Appendix E contains the details of our architecture and hyperparameters necessary to reproduce our results. Also, our code is available at `https://github.com/FER-multifrequency-so3/FER-multifrequency-so3`.

ACKNOWLEDGMENTS

This work was supported by Institute of Information & communications Technology Planning & Evaluation (IITP) grant funded by the Korea government (MSIT) (No.2019-0-00075, Artificial Intelligence Graduate School Program (KAIST)), (No.2022-0-00311, Development of Goal-Oriented Reinforcement Learning Techniques for Contact-Rich Robotic Manipulation of Everyday Objects), (No. 2022-0-00612, Geometric and Physical Commonsense Reasoning based Behavior Intelligence for Embodied AI).

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

## A    PROOF FOR THEOREM 1 IN THE MAIN PAPER

**Theorem 1.** *If $J_i \in \mathbb{R}^{n \times n} \; \forall i \in \{1, 2, 3\}$ satisfies $-J_i = J_i^T$, $[J_1, J_2] = J_3$, $[J_2, J_3] = J_1$, $[J_3, J_1] = J_2$ where $[A, B] = AB - BA$, and $\exp(2m\pi J_i) = I_{n \times n}, \forall m \in \mathbb{Z}$, where $\mathbb{Z}$ is the space of integers, then $D(R) = \exp(\theta \hat{w} \cdot \vec{J}) \in SO(n)$ and satisfies the conditions in Equation 2.*

In this section, the proof of Theorem 1 and Proposition 1 from the main paper is described. Here, the terms including $\vec{F} = [F_1, F_2, F_3]$, $\vec{J} = [J_1, J_2, J_3]$, $D(R)$ are all from Section 4.

$$\forall R_1, R_2 \in SO(3), \text{if } R_1 \neq R_2, \text{then } D(R_1) \neq D(R_2) \text{ (Lemma 8)}$$

$$\forall R \in SO(3), D \text{ maps } R \text{ to a single matrix } D(R) \in \mathbb{R}^{n \times n} \text{ (Lemma 1)}$$

$$\forall R \in SO(3), D(R)D(R)^T = D(R)^T D(R) = I \text{ (Lemma 9)}$$

$$\forall R \in SO(3), \det(D(R)) = 1 \text{ (Lemma 10)}$$

Additionally, we prove the compatibility of $D(R)$ which is necessary to prove the equivariance of $\psi(\vec{x})$.

$$D(R_1)D(R_2) = D(R_1 R_2) \text{ for all } R_1, R_2 \in SO(3) \text{ (Lemma 4)}$$

In the proposed solution of $D(R)$, the axis-angle representation is used. However, given a rotation matrix $R$, there are infinite ways to make a corresponding axis-angle representation. To be a high-dimensional rotation, there should be only one $D(R)$ corresponding to $R$. So, Lemma 1 shows the uniqueness of $D(R)$ given a single rotation matrix $R$.

**Lemma 1.** *Suppose $D$ satisfies the conditions in Equation 2, and $J_1, J_2, J_3$ satisfy the conditions in Theorem 1. $\forall R \in SO(3), D$ maps $R$ to a single matrix $D(R) \in \mathbb{R}^{n \times n}$.*

*Proof.* Let's assume that two different axis-angle representations $(\hat{\omega}_1, \theta_1), (\hat{\omega}_2, \theta_2)$ of $R$ are mapped into two different $D(R)$, which are $\exp(\theta_1 \hat{\omega}_1 \cdot \vec{J}) \neq \exp(\theta_2 \hat{\omega}_2 \cdot \vec{J})$.

$R = \exp(\theta_1 \hat{\omega}_1 \cdot \vec{F}) = \exp(\theta_2 \hat{\omega}_2 \cdot \vec{F})$ holds. From comparing each element of $\exp(\theta_1 \hat{\omega}_1 \cdot \vec{F}) = \exp(\theta_2 \hat{\omega}_2 \cdot \vec{F})$, $\exp(\theta_1 \hat{\omega}_1 \cdot \vec{J}) = \exp(\theta_2 \hat{\omega}_2 \cdot \vec{J})$ should be satisfied. So a contradiction has been reached in the assumption. $\square$

To show the compatibility of $D$, understanding the multiplication of two matrix exponentials is important because $D$ is defined as the matrix exponential. *Baker–Campbell–Hausdorff formula* is an expression of solution for $Z$ to the equation $e^X e^Y = e^Z$, which is described in Lemma 2.

**Lemma 2.** *(Baker–Campbell–Hausdorff formula)*

$$\exp(X)\exp(Y) = \exp(Z)$$

*where $X, Y \in \mathbb{R}^{n \times n}$ and the commutator $[\cdot, \cdot]$ is defined as $[X, Y] = XY - YX$.*

*The solution for $Z$ to the equation is as a series in $X$ and $Y$ and iterated commutator thereof. The first few terms of this series are*

$$Z = X + Y + \frac{1}{2}[X, Y] + \frac{1}{12}[X, [X, Y] - \frac{1}{12}[Y, [X, Y] + \cdots$$

The important thing here is that the formula is only affected by the commutator, and not affected by the dimension of the matrix. Lemma 3 says that if a set of three matrices satisfy a cyclic commutator condition, then the solution for $Z$ is a linear combination of these three matrices and their coefficients are independent of the dimension of the matrix. By using the fact that $\vec{F} = [F_1, F_2, F_3]$ and $\vec{J} = [J_1, J_2, J_3]$ both satisfies the cyclic commutator condition of Lemma 3.

**Lemma 3.** *Given a set of matrices*

$$\{A_1, A_2, A_3\} \subset \mathbb{R}^{m \times m} (m \in \mathbb{N} \geq 3)$$

*that satisfies*

$$[A_1, A_2] = A_3, [A_2, A_3] = A_1, [A_3, A_1] = A_2, where\ [X, Y] := XY - YX$$

$$\vec{a} \cdot \vec{A} = (a_1 A_1 + a_2 A_2 + a_3 A_3), \vec{a} \in \mathbb{R}^3$$

$$\exp(\vec{a} \cdot \vec{A}) \exp(\vec{b} \cdot \vec{A}) = \exp(z) \quad s.t. \quad z \in \mathbb{R}^{m \times m}$$

*Then*

$$z = g(\vec{a}, \vec{b}) \cdot \vec{A}$$

*where*

$$g : \mathbb{R}^3 \times \mathbb{R}^3 \to \mathbb{R}^3$$

*is independent of the dimension of the matrices* $m$

*Proof.* $\forall \vec{a}, \vec{b} \in \mathbb{R}^3$, we can define a function $f : \mathbb{R}^3 \times \mathbb{R}^3 \to \mathbb{R}^3$ where $f(\vec{a}, \vec{b}) = [a_2 b_3 - a_3 b_2, a_3 b_1 - a_1 b_3, a_1 b_2 - a_2 b_1]$

$$[\vec{a} \cdot \vec{A}, \vec{b} \cdot \vec{A}] = (a_1 b_2 - a_2 b_1)A_3 + (a_2 b_3 - a_3 b_2)A_1 + (a_3 b_1 - a_1 b_3)A_2$$
$$= f(\vec{a}, \vec{b}) \cdot \vec{A}$$

The important thing here is that $f$ is independent of the dimension of the matrices $m$.

*let* $\exp(\vec{a} \cdot \vec{A}) \exp(\vec{b} \cdot \vec{A}) = \exp(z) \quad$ s.t. $\quad z \in \mathbb{R}^{m \times m}$

$$z = \vec{a} \cdot \vec{A} + \vec{b} \cdot \vec{A} + \frac{1}{2}[\vec{a} \cdot \vec{A}, \vec{b} \cdot \vec{A}] + \frac{1}{12}[\vec{a} \cdot \vec{A}, [\vec{a} \cdot \vec{A}, \vec{b} \cdot \vec{A}]] + \cdots$$
$$= (\vec{a} + \vec{b} + \frac{1}{2}f(\vec{a}, \vec{b}) + \frac{1}{12}f(\vec{a}, f(\vec{a}, \vec{b})) + \cdots) \cdot \vec{A}$$
$$= g(\vec{a}, \vec{b}) \cdot \vec{A}$$

Since every term of $z$ consists of iterative commutators, the result of every iterative commutator will be $\vec{c} \cdot \vec{A}$ where $\vec{c} \in \mathbb{R}^3$. Since $\vec{c}$ is only related to $f$, $g : \mathbb{R}^3 \times \mathbb{R}^3 \to \mathbb{R}^3$ is independent of $m$.

$\square$

**Lemma 4.** *Suppose $D$ satisfies the conditions in Equation 2, and $J_1, J_2, J_3$ satisfy the conditions in Theorem 1. $D(R_1)D(R_2) = D(R_1 R_2)$ for all $R_1, R_2 \in SO(3)$*

*Proof.* For any rotation matrix $R \in SO(3)$, there exist an axis-angle representation with the axis $\hat{\omega}$ and angle $\theta$,

$$R = \exp(\theta \hat{\omega} \cdot \vec{F})$$

where $\{F_1, F_2, F_3\}$ is the basis for $3 \times 3$ skew-symmetric matrix defined in the prerequisites, and satisfies Lemma 3 with a function $g$.

$$\forall \vec{a}, \vec{b} \in \mathbb{R}^3, \exp(\vec{a} \cdot \vec{F}) \exp(\vec{b} \cdot \vec{F}) = \exp(g(\vec{a}, \vec{b}) \cdot \vec{F})$$

Also, our proposed high-dimensional rotation $D$

$$D(R) = \exp(\theta \hat{\omega} \cdot \vec{J})$$

where $\{J_1, J_2, J_3\}$ satisfies Lemma 3 with a function $g$.

$$\forall \vec{a}, \vec{b} \in \mathbb{R}^3, \exp(\vec{a} \cdot \vec{J}) \exp(\vec{b} \cdot \vec{J}) = \exp(g(\vec{a}, \vec{b}) \cdot \vec{J})$$

Since function $g$ from Lemma 3 is independent of the dimension of matrices, the function $g$ used in $R$ and $D$ are identical.

Since $R_1 R_2 \in SO(3)$, we can get axis-angle representations that are axes $\hat{\omega}_1, \hat{\omega}_2, \hat{\omega}_3 \in \mathbb{R}^3$ and angles $\theta_1, \theta_2, \theta_3$ of each $R_1, R_2$, and $R_1 R_2$.

$$R_1 = \exp(\theta_1 \hat{\omega}_1 \cdot \vec{F}), R_2 = \exp(\theta_2 \hat{\omega}_2 \cdot \vec{F}), R_1 R_2 = \exp(\theta_3 \hat{\omega}_3 \cdot \vec{F})$$

$$R_1 R_2 = \exp(\theta_1 \hat{\omega}_1 \cdot \vec{F}) \exp(\theta_2 \hat{\omega}_2 \cdot \vec{F}) = \exp(g(\theta_1 \hat{\omega}_1, \theta_2 \hat{\omega}_2) \cdot \vec{F}) = \exp(\theta_3 \hat{\omega}_3 \cdot \vec{F})$$

From Lemma 1,

$$\exp(g(\theta_1 \hat{\omega}_1, \theta_2 \hat{\omega}_2) \cdot \vec{J}) = \exp(\theta_3 \hat{\omega}_3 \cdot \vec{J})$$

$$\therefore D(R_1)D(R_2) = \exp(\theta_1 \hat{\omega}_1 \cdot \vec{J}) \exp(\theta_2 \hat{\omega}_2 \cdot \vec{J})$$

$$= \exp(g(\theta_1 \hat{\omega}_1, \theta_2 \hat{\omega}_2) \cdot \vec{J}) = \exp(\theta_3 \hat{\omega}_3 \cdot \vec{J}) = D(R_1 R_2)$$

$$\square$$

To prove the injectiveness of $D$, we will first show that $J_1, J_2, J_3$ share n eigenvalues in Lemma 5. By using this, we will prove Lemma 8, which means that $D$ is injective.

**Lemma 5.** *If $J_1, J_2, J_3 \in \mathbb{R}^{n \times n}$ that satisfies commutator relationship $[J_i, J_j] = J_k$ where $(i, j, k)$ are cyclic permutations of $(1, 2, 3)$, they share the same eigenvalues including both the values and their multiplicities.*

*Proof.* Without loss of generality, we will show there exists a similarity transform between $J_1$ and $J_2$. With a scalar $\theta \in \mathbb{R}$

$$\exp(\theta J_3) J_1 \exp(-\theta J_3) = (I + \theta J_3 + \frac{1}{2!}\theta^2 J_3^2 + \cdots) J_1 (I - \theta J_3 + \frac{1}{2!}\theta^2 J_3^2 - \cdots)$$

$$= J_1 + \theta(J_3 J_1 - J_1 J_3) + \frac{\theta^2}{2!}(J_3^2 J - 1 - 2J_3 J_1 J_3 + J_1 J_3^2) + \cdots$$

$$= J_1 + \theta[J_3, J_1] + \frac{\theta^2}{2!}[J_3, [J_3, J_1]] + \frac{\theta^3}{3!}[J_3, [J_3, [J_3, J_1]]] + \cdots$$

$$= J_1\{1 - \frac{\theta^2}{2!} + \cdots\} + J_2\{\theta - \frac{\theta^3}{3!} + \cdots\}$$

$$= J_1 \cos\theta + J_2 \sin\theta$$

So, if $\theta = \pi/2$, $\exp(\theta J_3) J_1 \exp(-\theta J_3) = J_2$ holds. Since $\exp(\theta J_3) \exp(-\theta J_3) = I$, There exists a similarity transform between $J_1$ and $J_2$. So, $J_1$ and $J_2$ share the eigenvalues and multiplicities. By doing the same way, it can be proved that $J_1, J_2$ and $J_3$ share the same eigenvalues and multiplicities.

$$\square$$

In Lemma 6, $J_+$ and $J_-$ is defined. They are used in Proposition 1 to find the eigenvalues of $J_3$, since if there exists an eigenvalue $\lambda$ of $J_3$ s.t. $J_3 \vec{v} = \lambda \vec{v}$, then $J_3 J_- \vec{v} = (\lambda + i)J_- \vec{v}$ and $J_3 J_+ \vec{v} = (\lambda - i)J_+ \vec{v}$ hold. So applying $J_-, J_+$ makes the eigenvalue one step bigger or smaller, or the corresponding eigenvector is in the nullspace of $J_-, J_+$.

**Lemma 6.** *Suppose $J_1, J_2, J_3$ satisfy the conditions in Theorem 1. Define*

$$J_+ = J_1 + iJ_2$$

$$J_- = J_1 - iJ_2$$

*Then $J_+, J_-$ share the same eigenvalues include multiplicity with $J_i$ and satisfies $[J_3, J_\pm] = -iJ_\pm$*

*Proof.* By proposition 2, eigenvalues of $J_1$ and $J_2$ are the same, with n eigenvalues. If $\lambda_1$ is an eigenvalue of $J_1$ and $J_2$ and there will be corresponding eigenvectors $v_1, v_2 \in \mathbb{R}n$ for each $J_1, J_2$.

$$J_1 \vec{v_1} = \lambda_1 \vec{v_1}$$

$$J_2 \vec{v_2} = \lambda_1 \vec{v_2}$$

$$J_+ \lambda_1 = (J_1 + iJ_2)\lambda_1 = (\vec{v_1} + i\vec{v_2})\lambda_1$$

$$J_- \lambda_1 = (J_1 - iJ_2)\lambda_1 = (\vec{v_1} - i\vec{v_2})\lambda_1$$

$\lambda_1$ is also an eigenvalue of $J_+, J_-$. So, $J_+, J_-$ share the same eigenvalues with $J_i$. The same procedure can be applied when $\lambda_1$ has a multiplicity bigger than 1.

Also, by applying commutator on $J_\pm$,

$$[J_3, J_\pm] = [J_3, (J_1 \pm iJ_2)] = [J_3, J_1] \pm i[J_3, J_2]$$
$$= J_2 \mp iJ_1 = \mp iJ_\pm$$

$\square$

**Proposition 1.** *Suppose $J_1, J_2, J_3$ satisfy the conditions in Theorem 1. Then, $J_1, J_2$, and $J_3$ have the same eigenvalues and multiplicities. In particular, the eigenvalues are $\Lambda = \{-k\mathrm{i}, -(k-1)\mathrm{i}, \ldots, -\mathrm{i}, 0, \mathrm{i}, \ldots, k\mathrm{i}\}$ for some non-negative integer $k$. Further, if $\lambda$ is an eigenvalue of $J_i$ with multiplicity $m$, then $-\lambda$ is also an eigenvalue of $J_i$ with the same multiplicity $m$.*

*Proof.* **Sharing eigenvalues** $J_1, J_2$, and , $J_3$ have same eigenvalues and multiplicities, proved by Lemma 5.

**Integer eigenvalue coefficients.** Since $J_i \in \{J_1, J_2, J_3\}$ is a real skew-symmetric matrix, it can be diagonalized into $J_i = P(\Lambda)P^{-1}$ where $P \in \mathbb{C}^{n \times n}$ is a matrix and $\Lambda \in \mathbb{C}^{n \times n}$ is diagonal matrix whose diagonal elements $\lambda_i$ are the eigenvalues of $J_i$.

Then, from the condition of $J_i : \exp(2k\pi J_i) = I_{n \times n}, \forall i \in \{1, 2, 3\}$

$$\exp(2k\pi J_i) = P \begin{bmatrix} \exp(2k\pi\lambda_1) & \cdots & 0 \\ \vdots & \ddots & \vdots \\ 0 & \cdots & \exp(2k\pi\lambda_n) \end{bmatrix} P^{-1} = I_{n \times n}$$

$$\begin{bmatrix} \exp(2k\pi\lambda_1) & \cdots & 0 \\ \vdots & \ddots & \vdots \\ 0 & \cdots & \exp(2k\pi\lambda_n) \end{bmatrix} = P^{-1} I_{n \times n} P = I_{n \times n}$$

So, $\exp(2k\pi\lambda_i) = 1$ should be satisfied for $\forall k \in \mathbb{Z}, \forall i \in \{1, 2, \ldots, n\}$. Therefore,

$$\lambda_i = m\mathrm{i}, \exists m \in \mathbb{Z}$$

**Existence of null vector.** From Lemma 5 and 6, $J_1, J_2, J_3, J_+, J_-$ shares the eigenvalues. Let's choose one eigenvalue $\lambda_i$ and the corresponding eigenvector $v_i$ of $J_3$. Then,

$$J_3 v_i = \lambda_i v_i$$
$$J_3 J_- v_i = ([J_3, J_-] + J_- J_3) v_i = iJ_- v_i + J_-(\lambda_i v_i) = (\lambda_i + i) J_- v_i$$

Here if $J_- v_i \neq \vec{0}$ then $\lambda_i + i$ is eigenvalue of $J_3$ and $J_- v_i$ is the corresponding eigenvector. Then, by doing the same way, if $J_-(J_- v_i) \neq \vec{0}$, then $\lambda_i + 2i$ is eigenvalue of $J_3$. This procedure can be repeated, but since $J_3$ is a finite matrix, its eigenvalue is also finite. So we can always find $J_-(J_-^{(m)} v_i) = \vec{0}$ where $J_-^{(m)} v_i \neq \vec{0}$ .

**Consecutive eigenvalues** Then, let's define $m$ as the dimension of the null space of $J_3$. Then we can always find the real orthonormal basis $v_1, v_2, \cdots, v_m$ of the null space of $J_3$ since $J_3$ is a real matrix. Let's choose $v_j \in \mathbb{R}^n$ and it satisfies $J_3 v_j = \vec{0}$. Then if $J_- v_j \neq \vec{0}$ then $i$ is eigenvalue of $J_3$ and $J_- v_j$ is the corresponding eigenvector.

$$J_3(J_- v_j) = iJ_3$$

Apply conjugate: $J_3(J_+ v_j) = -iJ_3$

So $-i$ is also an eigenvalue of $J_3$ and $J_+ v_j$ is the corresponding eigenvector. By applying this procedure repeatedly, we can find the first $n_j \geq 0 \in \mathbb{Z}$ that satisfies $J_-^{(n_j+1)} v_j = \vec{0}$, and we can find the continuous interval of eigenvalues

$$\Lambda_j = \{-n_j i, \cdots, -i, 0, i, \cdots, n_j i\}$$

with corresponding eigenvectors

$$\{J_+^{(n_j)}v_j, \cdots, J_+ v_j, v_j, J_- v_j, \cdots, J_-^{(n_j)}v_j\}$$

**Completeness** For each $\Lambda_j$, we can find $J_-^{(n_j+1)}v_j = J_-(J_-^{(n_j)}v_j) = \vec{0}$ where $J_-^{(n_j)}v_j \neq \vec{0}$. So $J_-^{(n_j)}v_j$ is a nonzero null vector of $J_-$ while also an eigenvector of $J_3$ corresponding to the eigenvalue $n_j i$.

If there is another nonzero eigenvalue $\lambda_x$ outside of $\Lambda_1 + \Lambda_2 + ... + \Lambda_m$ and corresponding eigenvector $v_x$, then $v_x$ is independent with $v_1, v_2, \cdots, v_m$. We can always find $J_-^{(n_x)}v_x$, which is a nonzero null vector of $J_-$ while also an eigenvector of $J_3$ corresponding to the eigenvalue $\lambda_x + n_x i$. So we found a nonzero null vector of $J_-$ that is independent with existing $k$ null vectors of $J_-$, and it makes the dimension of null space of $J_-$ is bigger than $k$. However it is a contradiction since the dimension of null space of $J_3$ is $k$ and $J_-$ shares the eigenvalues with $J_3$.

$\square$

**Lemma 7.** *Suppose $D$ satisfies the conditions in Equation 2, and $J_1, J_2, J_3$ satisfy the conditions in Theorem 1.*

$$D(R^{-1}) = D(R)^{-1} \text{ for any } R \in SO(3)$$

*Proof.* From Lemma 4,

$$D(R)D(R^{-1}) = D(RR^{-1}) = I_{n\times n}, D(R^{-1})D(R) = D(R^{-1}R) = I_{n\times n}$$

$$D(R^{-1}) = D(R)^{-1}$$

$\square$

**Lemma 8.** *Suppose $D$ satisfies the conditions in Equation 2, and $J_1, J_2, J_3$ satisfy the conditions in Theorem 1. Then, $\forall R_1, R_2 \in SO(3), if R_1 \neq R_2, then D(R_1) \neq D(R_2)$*

*Proof.* Let's prove a rotation matrix which is mapped to $I_{n\times n}$ is only $I_{3\times 3}$

Let's assume that there exists another rotation matrix $R \neq I_{3\times 3}$ which is mapped to $I_{n\times n}$. Then we can find an axis-angle representation of $R$ which is the axis $\hat{\omega} = [w_1, w_2, w_3]$ and the angle $\theta \neq 0$. Let's define $J = \hat{\omega} \cdot \vec{J}$. Then $D(R) = \exp(\theta J) = I_{n\times n}$.

From Proposition 1, eigenvalues of $J$ is $\Lambda = \{-ki, \cdots, -i, 0, i, \cdots, ki\}, k \in \mathbb{Z}$ including multiplicities. Then eigenvalues of $\exp(\theta J)$ is $\{e^{\theta\lambda_i i}\}$ where $\lambda_i \in \Lambda$. The eigenvalue of $I_{n\times n}$ is only 1, so every $e^{\theta\lambda_i i}$ should be 1. Then $\theta$ should be $2k\pi$, but it makes $R = I_{3\times 3}$, so it is contradiction.

If there is two distinct $R_1 \neq R_2$ mapped to same $\mathbb{R}^{n\times n}$, then $D(R_1 R_2^{-1}) = D(R_1)D(R_2^{-1}) = D(R_1)D(R_2)^{-1} = I_{n\times n}$ from Lemma 7. But $R_1 R_2^{-1}$ is not $I$, so there is contradiction. Therefore $D$ is injective.

$\square$

**Lemma 9.** *Suppose $D$ satisfies the conditions in Equation 2, and $J_1, J_2, J_3$ satisfy the conditions in Theorem 1. Then, $\forall R \in SO(3), D(R)D(R)^T = D(R)^T D(R) = I$*

*Proof.*

$$\exp(\vec{0} \cdot \vec{F}) = \exp(\vec{0}) = I_{3\times 3} \in \mathbb{R}^3$$

So $\vec{0}$ is obviously one possible axis-angle representation of $I_{3\times 3}$, and $D(I_{3\times 3}) = \exp(\vec{0} \cdot \vec{J}) = I_{n\times n}$. From Lemma 1, $D(I_{3\times 3}) = I_{n\times n}$ is unique. $\square$

**Lemma 10.** *Suppose $D$ satisfies the conditions in Equation 2, and $J_1, J_2, J_3$ satisfy the conditions in Theorem 1. $\forall R \in SO(3), det(D(R)) = 1$*

*Proof.*

$$1 = det(D(RR^T)) = det(D(R)D(R^T)) = det(D(R))det(D(R^T)) = det(D(R))^2$$

Therefore, $det(D(R)) = 1$ or $-1 \ \forall R \in SO(3)$.

Let's assume $det(D(R)) = -1$ for all $R \in SO(3)$. Then, for $R_1, R_2 \in SO(3)$,

$$1 = det(D(R_1))det(D(R_2)) = det(D(R_1 R_2)) = -1$$

which is contradictory. Therefore $det(D(R)) = 1$ for all $R \in SO(3)$.

$\square$

## B  PROOF OF THEOREM 2 IN THE MAIN PAPER

**Theorem 2.** *Suppose $D$ satisfies the conditions in Equation 2, and $J_1$, $J_2$, $J_3$ satisfy the conditions in Theorem 1. $\forall R \in SO(3)$ whose angle of rotation is $\theta$ and rotation axis is $\hat{\omega}$, we have*

$$D(R) = \exp(\theta\hat{\omega} \cdot \vec{J}) = \sum_{\lambda \in \Lambda} \vec{b}_\lambda \vec{b}_\lambda^{*T}(\sin(\lambda\theta) + \mathrm{i}\cos(\lambda\theta))$$

*where $\vec{b}_\lambda$ is the eigenvector of $\theta\hat{\omega} \cdot \vec{J}$ that corresponds to eigenvalue $\lambda$.*

*Proof.* Since $J_i$ and $\hat{\omega} \cdot \vec{J}$ are real skew-symmetric matrices, they are diagonalizable. Further, by proposition 2, $\hat{\omega} \cdot \vec{J}$ has the same eigenvalues as $J_i$. This allows us to write $\exp(\theta\hat{\omega} \cdot \vec{J}) = \exp(P(\Lambda)P^{-1}) = P\exp(\Lambda)P^{-1}$ where $P \in \mathbb{C}^{n \times n}$ is a matrix and $\Lambda \in \mathbb{C}^{n \times n}$ is diagonal matrix of the eigenvalues of $\mathbf{J}$ multiplied by $\theta$. So, $D(R)$ can be expressed with

$$D(R) = P \begin{bmatrix} \exp(-ki\theta) & \cdots & 0 \\ \vdots & \ddots & \vdots \\ 0 & \cdots & \exp(ki\theta) \end{bmatrix} P^{-1} = P \begin{bmatrix} \cos(-k\theta)+\mathrm{i}\sin(-k\theta) & \cdots & 0 \\ \vdots & \ddots & \vdots \\ 0 & \cdots & \cos(k\theta)+\mathrm{i}\sin(k\theta) \end{bmatrix} P^{-1}$$

From this, we can write

$$D(R) = \exp(\theta\hat{\omega} \cdot \vec{J}) = P\Lambda P^{-1} = \sum_{i=1}^{n} \vec{b}_i \vec{b}_i^{*T}(\sin(k_i\theta) + \mathrm{i}\cos(k_i\theta))$$

where $\vec{b}_i$ is the $i^{th}$ eigenvector of $\hat{\omega} \cdot \vec{J}$.

$\square$

## C  PROOF OF THEOREM 3 IN THE MAIN PAPER

**Theorem 3.** *Consider the mapping $\psi(\vec{u}) = ||\vec{u}||D(R^z(\hat{u}))\hat{e}$. $\psi$ is rotational equivariant if $\hat{e}$ is the eigenvector corresponding to the zero eigenvalue of $J_3$, and $D$ satisfies all the conditions in equation 2.*

*Proof.* We recall the definition of $\psi$

$$\psi(\vec{x}) = \varphi(||\vec{x}||)D(R^z(\hat{x}))v$$

Based on this definition, the theorem would be true if

$$\psi(R\vec{x}) = \varphi(|R\vec{x}|)D(R^z(R\hat{x}))v$$
$$= \varphi(||\vec{x}||)D(R^z(R\hat{x}))v$$

is equivalent to

$$D(R)\psi(\vec{x}) = D(R)\varphi(||\vec{x}||)D(R^z(\hat{x}))v$$

By definition, $R^z(\hat{k})$ maps $\hat{z} = [0, 0, 1]$ to $\hat{k}$ for some unit vector $\hat{k}$. So, we have

$$R \underbrace{R^z(\hat{x}) \cdot \hat{z}}_{\hat{x}} = R\hat{x}$$

and

$$\underbrace{R^z(R\hat{x}) \cdot \hat{z}}_{R\hat{x}} = R\hat{x}$$

Because $RR^z(\hat{x}) \cdot \hat{z} = Rx = R^z(R\hat{x}) \cdot \hat{z}$,

$$[RR^z(\hat{x})]^T [R^z(R\hat{x})] \cdot \hat{z} = [RR^z(\hat{x})]^T [RR^z(\hat{x})] \cdot \hat{z} = \hat{z}$$

So $[RR^z(\hat{x})]^T [R^z(R\hat{x})]$ must be a rotation about the z-axis. This means that there exists some $\gamma \in \mathbb{R}$ where $[RR^z(\hat{x})]^T [R^z(R\hat{x})] = R_z(\gamma)$, and $R_z(\gamma)$ means rotating $\gamma$ angle about the z-axis.

So $[R^z(R\hat{x})] = [RR^z(\hat{x})]R_z(\gamma)$ holds. Considering $\psi(R\vec{x})$,

$$\begin{aligned}\psi(R\vec{x}) &= \varphi(||\vec{x}||)D(R^z(R\hat{x}))v \\ &= \varphi(||\vec{x}||)D(RR^z(\hat{x})R_z(\gamma))v \\ &= \varphi(||\vec{x}||)D(R)D(R^z(\hat{x}))D(R_z(\gamma))v\end{aligned}$$

since compatibility of $D(R)$ holds from Lemma 4.

From the definition of $D$, $D(R_z(\gamma)) = \exp(\gamma J_3)$ holds. Also, considering $v$ is the eigenvector of $J_3$ corresponding to the zero eigenvalue, $J_3 v = 0$. From the definition of the matrix exponential $\exp(\gamma J_3) = I + (\gamma J_3) + (\gamma J_3)^2/2 + \cdots$, $\exp(\gamma J_3)v = v$ satisfies. So

$$\begin{aligned}\psi(R\vec{x}) &= \varphi(||\vec{x}||)D(RR^z(\hat{x}))v \\ &= \varphi(||\vec{x}||)D(R)D(R^z(\hat{x}))v \\ &= D(R)\psi(\vec{x})\end{aligned}$$

$\square$

## D   PROOFS OF REQUIRED PROPOSITIONS

**Proposition 1.** *Suppose $J_1, J_2, J_3$ satisfy the conditions in Theorem 1. Then, $J_1, J_2$, and $J_3$ have the same eigenvalues and multiplicities. In particular, the eigenvalues are $\Lambda = \{-k\mathrm{i}, -(k-1)\mathrm{i}, \ldots, -\mathrm{i}, 0, \mathrm{i}, \ldots, k\mathrm{i}\}$ for some non-negative integer $k$. Further, if $\lambda$ is an eigenvalue of $J_i$ with multiplicity $m$, then $-\lambda$ is also an eigenvalue of $J_i$ with the same multiplicity $m$.*

*Proof.* **Sharing eigenvalues** $J_1, J_2$, and $, J_3$ have same eigenvalues and multiplicities, proved by Lemma 5.

**Integer eigenvalue coefficients.** Since $J_i \in \{J_1, J_2, J_3\}$ is a real skew-symmetric matrix, it can be diagonalized into $J_i = P(\Lambda)P^{-1}$ where $P \in \mathbb{C}^{n \times n}$ is a matrix and $\Lambda \in \mathbb{C}^{n \times n}$ is diagonal matrix whose diagonal elements $\lambda_i$ are the eigenvalues of $J_i$.

Then, from the condition of $J_i : \exp(2k\pi J_i) = I_{n \times n}, \forall i \in \{1, 2, 3\}$

$$\exp(2k\pi J_i) = P \begin{bmatrix} \exp(2k\pi\lambda_1) & \cdots & 0 \\ \vdots & \ddots & \vdots \\ 0 & \cdots & \exp(2k\pi\lambda_n) \end{bmatrix} P^{-1} = I_{n \times n}$$

$$\begin{bmatrix} \exp(2k\pi\lambda_1) & \cdots & 0 \\ \vdots & \ddots & \vdots \\ 0 & \cdots & \exp(2k\pi\lambda_n) \end{bmatrix} = P^{-1} I_{n \times n} P = I_{n \times n}$$

So, $\exp(2k\pi\lambda_i) = 1$ should be satisfied for $\forall k \in \mathbb{Z}, \forall i \in \{1, 2, \ldots, n\}$. Therefore,

$$\lambda_i = m\mathrm{i}, \exists m \in \mathbb{Z}$$

**Existence of null vector.** From Lemma 5 and 6, $J_1, J_2, J_3, J_+, J_-$ shares the eigenvalues. Let's choose one eigenvalue $\lambda_i$ and the corresponding eigenvector $v_i$ of $J_3$. Then,

$$J_3 v_i = \lambda_i v_i$$

$$J_3 J_- v_i = ([J_3, J_-] + J_- J_3)v_i = iJ_- v_i + J_-(\lambda_i v_i) = (\lambda_i + i)J_- v_i$$

Here if $J_- v_i \neq \vec{0}$ then $\lambda_i + i$ is eigenvalue of $J_3$ and $J_- v_i$ is the corresponding eigenvector. Then, by doing the same way, if $J_-(J_- v_i) \neq \vec{0}$, then $\lambda_i + 2i$ is eigenvalue of $J_3$. This procedure can be repeated, but since $J_3$ is a finite matrix, its eigenvalue is also finite. So we can always find $J_-(J_-^{(m)} v_i) = \vec{0}$ where $J_-^{(m)} v_i \neq \vec{0}$.

**Consecutive eigenvalues** Then, let's define $m$ as the dimension of the null space of $J_3$. Then we can always find the real orthonormal basis $v_1, v_2, \cdots, v_m$ of the null space of $J_3$ since $J_3$ is a real matrix. Let's choose $v_j \in \mathbb{R}^n$ and it satisfies $J_3 v_j = \vec{0}$. Then if $J_- v_j \neq \vec{0}$ then $i$ is eigenvalue of $J_3$ and $J_- v_j$ is the corresponding eigenvector.

$$J_3(J_- v_j) = iJ_3$$

$$\text{Apply conjugate: } J_3(J_+ v_j) = -iJ_3$$

So $-i$ is also an eigenvalue of $J_3$ and $J_+ v_j$ is the corresponding eigenvector. By applying this procedure repeatedly, we can find the first $n_j \geq 0 \in \mathbb{Z}$ that satisfies $J_-^{(n_j+1)} v_j = \vec{0}$, and we can find the continuous interval of eigenvalues

$$\Lambda_j = \{-n_j i, \cdots, -i, 0, i, \cdots, n_j i\}$$

with corresponding eigenvectors

$$\{J_+^{(n_j)} v_j, \cdots, J_+ v_j, v_j, J_- v_j, \cdots, J_-^{(n_j)} v_j\}$$

**Completeness** For each $\Lambda_j$, we can find $J_-^{(n_j+1)} v_j = J_-(J_-^{(n_j)} v_j) = \vec{0}$ where $J_-^{(n_j)} v_j \neq \vec{0}$. So $J_-^{(n_j)} v_j$ is a nonzero null vector of $J_-$ while also an eigenvector of $J_3$ corresponding to the eigenvalue $n_j i$.

If there is another nonzero eigenvalue $\lambda_x$ outside of $\Lambda_1 + \Lambda_2 + ... + \Lambda_m$ and corresponding eigenvector $v_x$, then $v_x$ is independent with $v_1, v_2, \cdots, v_m$. We can always find $J_-^{(n_x)} v_x$, which is a nonzero null vector of $J_-$ while also an eigenvector of $J_3$ corresponding to the eigenvalue $\lambda_x + n_x i$. So we found a nonzero null vector of $J_-$ that is independent of existing $k$ null vectors of $J_-$, and it makes the dimension of null space of $J_-$ is bigger than $k$. However, it is a contradiction since the dimension of null space of $J_3$ is $k$ and $J_-$ shares the eigenvalues with $J_3$.

$\square$

**Proposition 2.** *Suppose $J_1, J_2, J_3$ satisfy the conditions in Theorem 1. Then, $\theta\hat{\omega} \cdot \vec{J}$ have the same eigenvalues as $J_i$, $\Lambda = \{-k\theta i, -(k-1)\theta i, \ldots, -\theta i, 0, \theta i, \ldots, k\theta i\}$*

*Proof.* Given a rotation $R \in SO(3)$, we can always find an euler angle representation $\alpha, \beta, \gamma$ where $\mathbb{R} = R_x(\alpha)R_y(\beta)R_z(\gamma)$, $R_k$ means the rotation along k-axis, and $\alpha, \beta, \gamma \in \mathbb{R}$.

Let's think about $R_z(\gamma)$ first. From Lemma 5,

$$\exp(\gamma J_3) J_1 \exp(-\gamma J_3) = J_1 \cos\gamma + J_2 \sin\gamma = (R_z(\gamma)[1, 0, 0]^T) \cdot \vec{J}$$

$$\exp(\gamma J_3) J_2 \exp(-\gamma J_3) = J_2 \cos\gamma - J_1 \sin\gamma = (R_z(\gamma)[0, 1, 0]^T) \cdot \vec{J}$$

$$\exp(\gamma J_3) J_3 \exp(-\gamma J_3) = J_3 = (R_z(\gamma)[0, 0, 1]^T) \cdot \vec{J}$$

By using these terms,

$$\exp(\gamma J_3)(\vec{\omega} \cdot \vec{J}) \exp(-\gamma J_3) = (R_z(\gamma)\vec{\omega}) \cdot \vec{J}$$

Here $\exp(\gamma J_3) = \exp([0, 0, \gamma] \cdot \vec{J}) = D(R_z(\gamma))$ means the rotation of $\gamma$ angle on $z$-axis.

By doing the same thing on the x-axis and y-axis, we can get below.

$$\exp(\alpha J_1)\exp(\beta J_2)\exp(\gamma J_3)(\vec{\omega}\cdot\vec{J})\exp(-\gamma J_3)\exp(-\beta J_2)\exp(-\alpha J_1)$$

$$= (R_x(\alpha)R_y(\beta)R_z(\gamma)\vec{\omega})\cdot\vec{J}$$

Since compatibility holds in $D(R)$, $D(R) = D(R_x(\alpha)R_y(\beta)R_z(\gamma)) = \exp(\alpha J_1)\exp(\beta J_2)\exp(\gamma J_3)$ So, $(R\vec{\omega})\cdot\vec{J} = D(R)\vec{\omega}\cdot\vec{J}D(R)^T$

Then, we can always find a rotation $R$ from $[0,0,1]$ to $\hat{\omega}$ and satisfies $\hat{\omega}\cdot\vec{J} = D(R)J_3 D(R)^T$. So $\hat{\omega}\cdot\vec{J}$ shares eigenvalues with $J_3$. From Proposition 1, the eigenvalues of $J_3$ are $\{-ki, -(k-1)i, \ldots, -i, 0, i, \ldots, ki\}$. Multiplying $\theta$ to a matrix also makes the eigenvalues to be multiplied by $\theta$, so the eigenvalues of $\theta\hat{\omega}\cdot\vec{J}$ are $\{-k\theta i, -(k-1)\theta i, \ldots, -\theta i, 0, \theta i, \ldots, k\theta i\}$. $\square$

**Proposition 3.** *The maximum value of $k$ in Proposition 2 is $\lfloor\frac{n-1}{2}\rfloor$.*

*Proof.* From Proposition 1, $J_3 \in \mathbb{R}^{n\times n}$ always have the eigenvalues of $\{-ki, -(k-1)i, \ldots, -i, 0, i, \ldots, ki\}$ For $k$ to be the maximum, the multiplicity should be 1 except zero eigenvalue, and the multiplicity of zero eigenvalue is 1 when $n$ is even, and 2 when $n$ is odd. So, the maximum value of $k$ is $\lfloor\frac{n-1}{2}\rfloor$. $\square$

**Proposition 4.** *If a skew-symmetric matrix $A \in \mathbb{R}^{n\times n}$ s.t. $A^T = -A$ has the eigenvalues $\lambda_i = m_i i$ where $i = 1, 2, \ldots, n$ and $m_i \in \mathbb{Z}$, then $\exp(2k\pi A) = I_{n\times n}$ where $k \in \mathbb{Z}$.*

*Proof.* Since A is a real skew-symmetric matrix, it can be diagonalized into $A = P(\mathbf{\Lambda})P^{-1}$ where $P \in \mathbb{C}^{n\times n}$ is a matrix and $\mathbf{\Lambda} \in \mathbb{C}^{n\times n}$ is a diagonal matrix whose diagonal elements $\lambda_i = m_i i$ are the eigenvalues of $A$.

$$\exp(2k\pi A) = P\begin{bmatrix} \exp(2m_1 k\pi i) & \cdots & 0 \\ \vdots & \ddots & \vdots \\ 0 & \cdots & \exp(2m_n k\pi i) \end{bmatrix}P^{-1} = PI_{n\times n}P^{-1} = I_{n\times n}$$

$\square$

# E ARCHITECTURE DETAILS

## E.1 LAYERS

In this section, we clarify the difference between the Vector Neurons and introduce new layers. We built our architecture on Vector Neuron's implementation. For all kinds of layers, the main difference is in the dimension of the vector-list feature $\mathbf{V}$. In Vector Neurons, $\mathbf{V} \in \mathbb{R}^{C\times 3}$. In our case, we employ an augmented $\mathbf{V} \in \mathbb{R}^{C\times(3+n)}$, where $C$ is the channel count and $3 + n$ is the dimension of an SO(3)-equivariant vector with an added feature size of $n$. This easily adapts to the existing SO(3)-equivariant layers of the Vector Neurons. A linear layer, for example, is redefined only by changing the vector dimension 3 to $3 + n$:

$$\mathbf{V}' = f_{lin}(\mathbf{V};\mathbf{W}) = \mathbf{W}\mathbf{V} \in \mathbb{R}^{C'\times(3+n)}$$

It is worth noting that number of learnable parameter is the same with original VN (i.e., number of elements in $\mathbf{W}$) while the proposed one achieve significantly better performance than that. We adopt the input edge convolution layer, linear layer, pooling layer, and invariant layer from Vector Neurons by simply changing the vector dimension. To this end, we also further introduce layers we devised for our method.

### E.1.1 FEATURE AUGMENTATION

For any input layer that is subject to the SO(3) transformation, we begin with the feature augmentation layer. Here, we define the concatenation of the original vector $\vec{u} \in \mathbb{R}^3$ and its high-dimensional representation $\psi(\vec{u}) \in \mathbb{R}^n$. Given multi-frequency feature dimensions such as 3+5 or 3+5+7, we iterate over each dimension and expand the feature by concatenating features with different frequencies:

$$\Psi(\vec{u}) = \oplus_{i=3,5,7\ldots} \psi_i(\vec{u})$$

where $\psi_i(\vec{u})$ is our feature representation given dimension of the feature $i$, which is constructed from $\vec{J}$ by Algorithm 1. Within our definition of $\psi$ in equation 1, we have $\varphi(\|\vec{u}\|)$ for the scale, and we use simple MLP with two fully connected layers of dimension 16 and ReLU. The input and output of this MLP is $\mathbb{R}$ to adjust the scale factor for each point.

### E.1.2 NON-LINEAR LAYERS

We introduce a new non-linear layer based on the magnitude scaling of a vector, which is SO(3) equivariant. We first take the Euclidean norm of each vector of the vector list feature $\mathbf{V}$ which gives us a vector $\vec{q} \in \mathbb{R}^C$. The i-th element $q_i$ is given by:

$$q_i = \|\mathbf{V}_{i,:}\|_2 \in \mathbb{R}, \quad i = 1, \ldots, C$$

The vector $\vec{q}$ is processed through a compact neural network featuring two MLP layers with bias, separated by a 2D ReLU activation layer, denoted as $f_\theta : \mathbb{R}^C \to \mathbb{R}^C$, $\vec{q} \mapsto \vec{q'}$. Subsequently, our non-linear layer is formulated as the element-wise scaling of each vector $\mathbf{V}_i$ by its corresponding $q'_i$, expressed as:

$$\mathbf{V}'_{i,j} = q'_i \cdot \mathbf{V}_{i,j} \quad \text{for } i = 1, \ldots, C \text{ and } j = 1, \ldots, (3+n)$$

### E.1.3 NORMALIZATION LAYERS

We found that the normalization layers introduced in Deng et al. (2021) do not help in improving the performance. We thus do not incorporate those layers in our architectures.

## E.2 ARCHITECTURAL DETAIL FOR EACH TASK

### E.2.1 POINT CLOUD CLASSIFICATION AND SEGMENTATION

We utilize PointNet (Qi et al. (2017a)) and DGCNN (Wang et al. (2019)) as the underlying architectures for both classification and segmentation tasks. Our approach seamlessly integrates with their existing Vector Neurons implementations. By incorporating an initial feature augmentation layer $\Psi(\vec{x}) \in \mathbb{R}^{(3+n)}$ and substituting the activation function as indicated in E.1, no further modifications to other layers are required. We utilize $n = 5$ for all models. We reduced our number of channels to $C/8$ where $C$ is the number of channels in the baseline models (Qi et al. (2017a), Wang et al. (2019)). This is more challenging than $C/3$ by Deng et al. (2021). We adopt all other hyperparameters from Deng et al. (2021).

### E.2.2 NEURAL IMPLICIT RECONSTRUCTION

In the Occupancy Network (Mescheder et al. (2019)), the encoder produces a latent code $z \in \mathbb{R}^C$ from the input point cloud $P \in \mathbb{R}^{N \times 3}$ and the decoder estimates the likelihood of a query point $\vec{x} \in \mathbb{R}^3$ being occupied, conditioned on this latent code $z$. Following the approach used in Deng et al. (2021), we substitute the standard PointNet encoder with its FER-VN-PointNet variant. This modified encoder outputs a vector-list feature $Z \in \mathbb{R}^{C' \times (3+n)}$. Our decoder function $\mathcal{O}$ is defined in others of three invariant compositions of high-dimensional features $\Psi(\vec{x}) \in \mathbb{R}^{(3+n)}, Z \in \mathbb{R}^{C' \times (3+n)}$:

$$\mathcal{O}(\vec{x}|Z) = f_\theta(\langle \Psi(\vec{x}), Z \rangle, \|\Psi(\vec{x})\|^2, \text{FER-VN-In}(Z)) \in [0, 1]$$

where FER-VN-In$(Z)$ is our FER-VN- variant of the invariant layer (section E.1) which maps SO(3) equivariant feature $Z \in \mathbb{R}^{C' \times (3+n)}$ to SO(3) invariant feature $\bar{z} \in \mathbb{R}^{C'}$. We use $C = 513$ for the Occupancy Network and $C' = 171$ for both VN-OccNet and FER-VN-OccNet following Deng et al. (2021) resulting in the same number of learnable parameters. We train the network for 300k iterations with a learning rate of 0.0001 and batch size of 64, selecting the models based on the best validation mIoU scores following Mescheder et al. (2019). We utilize $n = 5$ for all models.

### E.2.3 Normal Estimation

For this task, we leverage the VN-PointNet/DGCNN framework with an output head specifically tailored for point cloud normal estimation, as presented in Puny et al. (2021). Given that point cloud normal estimation is an SO(3)-equivariant task, we preserve the SO(3)-equivariance of features up to the final output layers. Specifically, we replace the invariant layer, max-pooling layer, and succeeding standard MLP layers from the architecture detailed in E.2.1. We instead incorporate four intermediate SO(3)-equivariant layers with activations. The concluding layer outputs a feature vector of size 1, represented as $\mathbf{n} \in \mathbb{R}^{N \times 1 \times 3}$, where $N$ denotes the input point cloud's number of points and 3 signifies each point's normal vector. However, due to our method's higher-dimensional features, the output dimension becomes $\mathbf{n} \in \mathbb{R}^{N \times 1 \times (3+n)}$. To align the output dimensions with those of the label, we employ a high-frequency representation of the ground-truth normal vectors $\mathbf{n}_{gt} \in \mathbb{R}^{N \times 3}$, denoted as $\hat{\mathbf{n}} = \Psi(\mathbf{n}_{gt}) \in \mathbb{R}^{N \times (3+n)}$ during the training. The loss function minimized is:

$$\frac{1}{N} \sum_{i=1}^{N} \left[ \left( 1 - \frac{\vec{n}_i}{||\vec{n_i}||} \cdot \hat{n}_i \right) + \min\left( ||\vec{n}_i - \hat{n}_i||^2, ||\vec{n}_i + \hat{n}_i||^2 \right) \right]$$

where $N$ is the total number of points in the point cloud, and $\vec{n}_i$ and $\hat{n}_i$ are the predicted and ground-truth high-frequency representation of normal vectors respectively for the $i$-th point in the input point cloud. The first term estimates the accuracy of the predicted normal direction, and the second term regularizes the length of the vector to 1. We adopt the default hyperparameters provided in Puny et al. (2021) including the number of channels. We utilize $n = 5$ for all models.

### E.2.4 Point cloud registration

Point cloud registration aims to align two sets of $N$ points $P_1, P_2 \in \mathbb{R}^{N \times 3}$ that come from the same shape. This is formally addressed by the Orthogonal Procrustes problem Schönemann (1966), which finds the analytical solution to a rotation matrix $R \in SO(3)$ that minimizes $\|P_1 R^T - P_2\|_F^2$. However, it requires the knowledge of the correspondences between the points in $P_1, P_2$ which is usually hard to achieve. Zhu et al. (2022) proposes a different approach, which solves for the latent codes $Z_1, Z_2 \in \mathbb{R}^{C \times 3}$ of Vector Neurons instead of the explicit point clouds when the correspondence between points is not known. Due to the initial layer's edge convolution in Vector Neurons, the channel dimension of $Z$ achieves permutation invariance, which facilitates correspondence-free feature registration. Following this approach, we utilize our encoder from the shape completion task (section 4.1) to find $R$ in the feature space. To acquire $R$ from the rotation in high-dimension $D(R) = \exp(\vec{\omega} \cdot \vec{\mathbf{J}})$ where $\vec{\omega} \in \mathbb{R}^3$, we solve the following optimization problem:

$$\vec{\omega} = \arg\min_{\vec{\omega}} \sum_{i \in \{3,5,7,\dots\}} \|\bar{Z}_{1;i}\{\exp(\vec{\omega} \cdot \vec{\mathbf{J}}_i)\}^T - \bar{Z}_{2;i}\|_F^2$$

where $\bar{Z}_{1;i}$ and $\bar{Z}_{2;i}$ are features corresponding to the augmented dimension $i \in \{3, 5, 7, \dots\}$ of high-dimensional features $Z_1, Z_2 \in \mathbb{R}^{C \times (3+n)}$. We use the Cross-Entropy Method (CEM) (De Boer et al. (2005)) as a solver.

## F Algorithmic Details to Get $J_1, J_2, J_3$

This section is for explaining details in Algorithm 1 and 2 to obtain $J_1, J_2, J_3$.

**Constraints.** In theorem 1, we have eight constraints to get $J_1, J_2, J_3$.

$$[J_1, J_2] = J_3 \tag{5}$$

$$[J_2, J_3] = J_1 \tag{6}$$

$$[J_3, J_1] = J_2 \tag{7}$$

$$J_1 + J_1^T = 0 \tag{8}$$

$$J_2 + J_2^T = 0 \tag{9}$$

$$J_3 + J_3^T = 0 \tag{10}$$

$$\exp(2\pi k J_1) = I \tag{11}$$

$$\exp(2\pi k J_2) = I \tag{12}$$

$$\exp(2\pi k J_3) = I \tag{13}$$

$$\text{Eigenvalue of } J_3 = \{-\lfloor \frac{n-1}{2} \rfloor \mathrm{i}, ..., \lfloor \frac{n-1}{2} \rfloor \mathrm{i}\} \tag{14}$$

These constraints are redundant, so we first analyze the dependency between conditions.

- equation 9 can be derived from equation 10, equation 8, and equation 7.
- equation 13 can be derived from equation 14 and other constraints with Lemma 5.
- equation 11 and equation 12 can be derived from Proposition 4.

Our strategy to get feasible $J_1, J_2, J_3$ consists of three steps.

1. Sample $J_3$ satisfying equation 10 and equation 14.
2. Construct linear bases of $J_1$ and $J_2$ with equation 6, equation 7, and equation 8.
3. Perform non-linear optimization with equation 5.

Here is a more detailed description of the procedures described in the paper. For step 2, we pick three equations (equation 6, equation 7, and equation 8) that can be converted into linear relations with elements of $J_1$ and $J_2$. This relation can be expressed as:

$$\underbrace{\begin{bmatrix} I_{n \times n} \otimes J_3 - J_3^T \otimes I_{n \times n} & -I_{n^2 \times n^2} \\ I_{n^2 \times n^2} & -J_3^T \otimes I_{n \times n} + I_{n \times n} \otimes J_3 \\ \frac{\partial(\text{VEC}(J_1) + \text{VEC}(J_1^T))}{\partial \text{VEC}(J_1)} & 0 \end{bmatrix}}_{A} \begin{bmatrix} \text{VEC}(J_1) \\ \text{VEC}(J_2) \end{bmatrix} = 0 \tag{15}$$

where $A \in \mathbb{R}^{3n^2 \times 2n^2}$, $\oplus$ denotes Kronecker product and $\text{VEC}(\cdot)$ is vectorization of matrix. We have $3n^2$ equations and $2n^2$ variables, so this can have no solution. But, we found out that there is redundancy within equations, where some equations are expressed by a linear combination of others. For example, $J_1 + J_1^T = 0$ have $n^2$ equations, but it contains both $J_1(i,j) + J_1(j,i) = 0$ and $J_1(j,i) + J_1(i,j) = 0$ where $J_1(i,j)$ means $i^{th}$ row and $j^{th}$ column elements of $J_1$. Additionally, we assume that there exists a solution satisfying all conditions in Theorem 1, so there should be a non-empty null space of $A$. Practically, we use EIG function in the Numpy library (Harris et al. (2020)) and always check the nullity of $A$ within Algorithm 1.

## G   ADDITIONAL EXPERIMENT

In this appendix, we delve into a series of experiments designed to validate the efficacy of our feature representation method across multiple contexts. By integrating our representation, we demonstrate notable enhancements in capturing intricate shape details with minimal impact on inference efficiency, particularly in shape compression tasks (G.2) and spherical shape regression (G.6). Throughout dimensional analysis to investigate how each dimensional feature contributes to the shape compression task(G.3), we get a clearer picture of what's going on. Finally, our experiments reveal that FER leads to heightened model robustness in tasks such as point cloud completion (G.1) and registration (G.5).

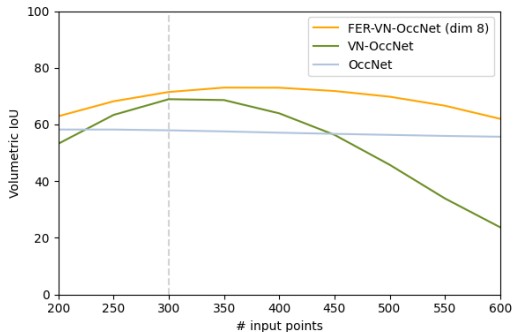

Figure 6: Volumetric mIoU on ShapeNet reconstruction with the different number of sampled input points. Every model is trained with the ShapeNet dataset where each data consists of 300 points.

### G.1 POINT CLOUD COMPLETION WITH DIFFERENT NUMBER OF SAMPLED POINTS

We experiment to assess the impact of the number of points. We train models with a dataset where each data consists of 300 points. We then evaluate the models using 200 to 600 input points during test time. As illustrated in the following figure 6, our method is more robust to changes in the number of points compared to both the Vector Neuron (VN)-based and vanilla Occupancy networks.

The reason why performance goes down as the number of points increases over 350 input points is that these methods extract point-wise features from a point cloud. VN uses K-Nearest-Neighbor(KNN) for each point-wise feature initialization, which processes the relative coordinates of adjacent points. Relative coordinates are sensitive to changes in point density. Such changes alter the scale of relative coordinates compared to what was observed during training. This change will decrease the performance not only when the number of points gets smaller but also larger.

### G.2 SHAPE COMPRESSION WITH DIFFERENT DIMENSIONALITY

We conduct additional experiments in the shape compression task to address the impact of the dimensionality of our feature representation on the computational cost and performance. Below are the results for processing 300 points for a single object for encoding and 100,000 query points for decoding. The results are averaged over 300 predictions.

| Model type | IoU (%) | inference time - encoder | inference time - decoder |
|---|---|---|---|
| VN-OccNet | 73.4 | 3.44 ± 0.05 ms | 9.57 ± 0.10 ms |
| FER-VN-OccNet (n=8) | 81.0 | 3.47 ± 0.06 ms | 9.64 ± 0.16 ms |
| FER-VN-OccNet (n=15) | 81.9 | 6.55 ± 0.26 ms | 9.87 ± 0.21 ms |

Table 6: Shape completion performance and inference time of encoder and decoder for VN-OccNet and FER-VN-OccNet with different dimensionality of feature representation.

As the table 6 shows, incorporating our 8-dimensional feature representation (FER-VN-OccNet (n=8)) boosts performance (a 7.6% increase in IoU) with a negligible impact on encoder inference time compared to the VN-OccNet baseline. However, expanding the feature representation to 15 dimensions (FER-VN-OCCNet (n=15)) doubles the computational time of the encoder with only a marginal performance improvement. On the other hand, the decoder inference time remained relatively stable across models, underscoring the efficiency of integrating FER into the network.

### G.3 SHAPE COMPRESSION OF EACH DIMENSIONAL FEATURE

We have an additional experiment on shape completion that compares which features are responsible for completing which part of the shape. Our composite feature is a concatenation of $n = 3, 5, 7, 9$ dimensional features (See Appendix E.1.1 for how we make the concatenation). The composite feature is then processed through a three-layer MLP to predict occupancy values in an occupancy network.

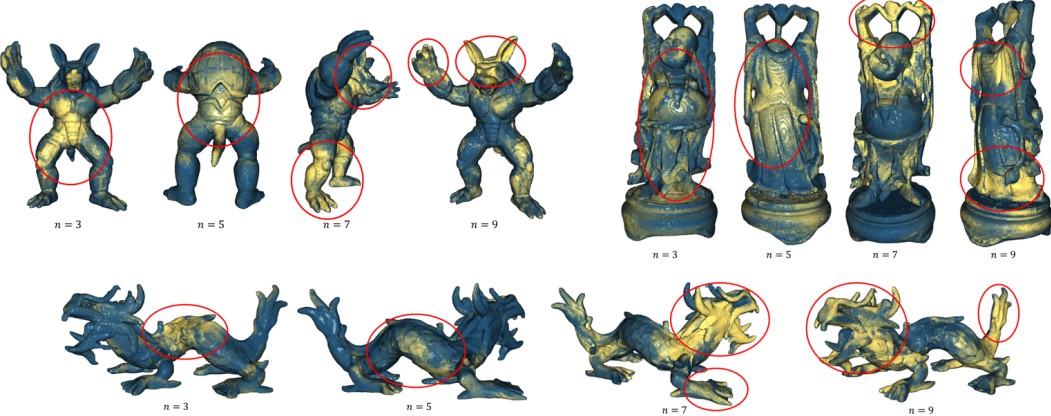

Figure 7: Shape compression result of our model by using the data from Stanford University (2023). $n$ means the dimensionality of features, and they are performed individually. The greater the magnitude of the gradient, the more yellow it is.

To visualize the contribution of each feature dimension on the completed shape, we compute the gradient magnitude of the occupancy predictions with respect to our feature representation of the surface points on the reconstructed mesh for each $n$. More concretely, we have $\psi_3(x), \psi_5(x), \psi_7(x), \psi_9(x)$ where $x \in \mathbb{R}^3$ is a surface point and $\psi_n(x)$ ($\mathbb{R}^3 \rightarrow \mathbb{R}^n$) is our feature presentation with $n$ number of dimensions. These four features are concatenated and processed by MLP to predict occupancy $o = f(\phi_3, \phi_5, \phi_7, \phi_9)$ where $f$ is MLP. Then, gradient used for visualization is $\partial o/\partial \phi_3$, $\partial o/\partial \psi_5$, $\partial o/\partial \psi_7$, $\partial o/\partial \psi_9$.

The visualization in figure 7 indicates that the lower-dimensional features, particularly for $n = 3$ and $n = 5$, predominantly capture the more expansive and volumetric components of the meshes, such as the torso and back of these objects. In contrast, the higher-dimensional features at $n = 7$ and $n = 9$ tend to focus on finer and more intricate details, such as facial features. These results offer valuable insights into the network's ability to differentiate between broad structural elements and detailed features of the meshes, reinforcing the effectiveness of frequency-based feature mappings in an SO(3) equivariant learning context.

### G.4    SHAPE COMPRESSION WITH IDENTICAL COMPRESSION RATIO

We conduct an additional experiment where we match the embedding size of all the methods, thus having the same compression ratio, and try to evaluate their reconstruction (i.e. decompression) performance. In this table 7, we observe that our method achieves the highest IoU while having the same compression ratio.

| Method | Input size | Embedding size | Compression ratio | IoU (%) |
|---|---|---|---|---|
| OccNet | 300 x 3 = 900 | 513 | 57% | 67.5 |
| VN-OccNet | 300 x 3 = 900 | 171 x 3 = 513 | 57% | 73.4 |
| FER-OccNet (n=3+5) | 300 x 3 = 900 | 64 x 8 = 512 | 57% | 77.3 |

Table 7: Shape completion performance measured by IoU for each model with the same compression ratio.

### G.5    POINT REGISTRATION WITH DIFFERENT INITIAL PERTURBATIONS

In the point registration application, the conventional Iterative Closest Points (ICP) method is known to be sensitive to initial guesses. That is, when the initial guess is far from the ground truth, ICP often fails to predict good results. On the other hand, the equivariance-based point registration method Zhu et al. (2022) is demonstrated for its robustness over initial guess. We hypothesize that the proposed FER-VN-based method also adopts the same characteristic, so we conduct the same

experiment in Zhu et al. (2022) to demonstrate this. To do so, in this experiment, we compare the performance of point registration for three different methods (FER-VN, VN, ICP) by changing the initial perturbation range.

| Max angle | 0 | 30 | 60 | 90 | 120 | 150 | 180 |
|---|---|---|---|---|---|---|---|
| FER-VN (ours) | 0.00348 | 0.00350 | 0.00349 | **0.00349** | **0.00350** | **0.00350** | **0.00349** |
| VN | 0.00649 | 0.00649 | 0.00649 | 0.00649 | 0.00649 | 0.00649 | 0.00649 |
| ICP | **0.00310** | **0.00312** | **0.00340** | 0.00508 | 0.00773 | 0.00966 | 0.01123 |
| GT | 0.00310 | | | | | | |

Table 8: Point registration performance measured by Chamfer Distance for each model by changing the range of initial perturbation. Max angle denotes the initial perturbation is sampled from a uniform distribution with bound $[0, \text{Max angle}]$ in degree.

As the table 8 demonstrates, our method is also robust to the initial perturbation, while ICP fails when we apply large perturbation. Notably, the performance with ours is significantly better than the one with VN and it almost reaches the performance of ground truth. Note that we add Gaussian noise to the input point clouds, so the ground truth CD is not zero.

### G.6 SPHERICAL SHAPE REGRESSION

To test the effectiveness of controlling the frequency of the feature representation, we employ a simple spherical shape regression experiment. We first pick a shape from EGAD (Morrison et al. (2020)). Then, we represent the shape's surface by changing the distance of points on a sphere. Basically, we train a model to predict how far a point is from the center when given a direction in 3D space. To train our model, we chose random directions and used ray casting to get the right distances for those directions.

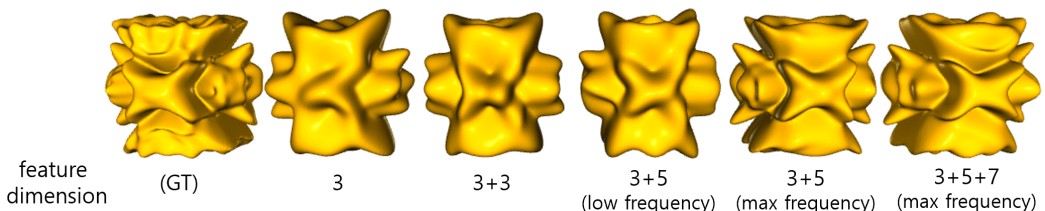

Figure 8: Toy experiment to show the effectiveness of controlling the frequency of the augmented features. One shape from EGAD (Morrison et al. (2020)) is regressed to networks based on the controlled features with different frequencies.

First, we train the model without feature augmentation. As the reconstructed shape labeled 3 in figure 8, the detained features are smooth out. When we increase the dimension of the representation by repeating the point coordinate twice to make the dimension of the representation 6 (3+3), still, the reconstruction quality is bad. This implies that just making the feature high-dimensional is not useful, and we need additional factors to increase expressiveness.

We observe how the frequency in the augmented feature is depicted by the eigenvalues of $J_i$. Assuming that utilizing only three dimensions does not adequately capture the shape details, we opt for the addition of 5-dimensional features. From Proposition 3, the eigenvalues including multiplicity will be either $\Lambda_1 = \{-i, 0, 0, 0, i\}$ or $\Lambda_2 = \{-2i, -i, 0, i, 2i\}$. Based on Theorem 2, higher magnitude eigenvalues render $\psi$ more sensitive to the input feature, thereby capturing more details. Fig. 8 demonstrates the impact of the frequency of the augmented feature on the shape details. A low frequency, where the 5-dimensional augmented feature used $J_3$ with eigenvalues of $\Lambda_1$, shows negligible improvement. However, employing the maximum frequency, with the augmented feature using $J_3$ having eigenvalues of $\Lambda_2$, results in a significant enhancement compared to the low-frequency augmented feature.

### G.7 APPLICATION OF FER ON VECTOR NEURON-BASED METHODS

Our feature representation, FER, can be integrated into any of VN-based methods. We applied our method to one of the recent VN-based methods, GraphONet Chen et al. (2022). We select 5 meshes from Stanford University (2023), and generate a dataset with 300 surface points and 1024 query points to evaluate occupancy. All meshes are normalized in scale, and we use query points generated by adding small Gaussian noise to the surface points. We train GraphONet baseline and FER-GraphONet, which is a variant of GraphONet by integrating our proposed feature representation (FER). We apply a feature dimension of 8 (3+5). To evaluate representation power, we adopt the test setting of the occupancy network, where reconstruction quality is evaluated on the meshes used in training. For the evaluation metric, we use volumetric intersection over union (IoU).

| Method | IoU (%) |
|---|---|
| GraphONet | 56.0 |
| FER-GraphONet | 57.6 |

Table 9: Shape reconstruction result measured by volumetric intersection over union (IoU). Our method is applied to GraphONet, and shows improved performance.

As shown in table 9, we observe enhanced performance, reinforcing the effectiveness of our approach. The result for GraphONet is 56.0% and 57.6% for FER-GraphONet, which shows 1.6% improvement. The results demonstrate that our proposed feature representation further improves performance when used with state-of-the-art methods.

