# OpenReview forum: "An Intuitive Multi-Frequency Feature Representation for SO(3)-Equivariant Networks"
_ICLR.cc/2024/Conference — ICLR 2024 poster_

### Official Review · Reviewer_HjVh · 2023-10-21

**Soundness:** 3 good
**Presentation:** 2 fair
**Contribution:** 3 good
**Rating:** 6
**Confidence:** 2

**Summary:**

**UPDATE**
After reading the rebuttal, I now better understand and appreciate the contribution. While I still believe this is a testimony to the much needed improvement in writing, I believe the work should be accepted.


The work proposes an SO(3) equivariant pointcloud network by extending vector neurons representation from a list of 3-dim vectors to a list of k-dim vectors. To accommodate this  increase in dimensionality a rotation operator construction is developed that can map a rotation in SO(3) to k dim.

**Strengths:**

- The work demonstrates improvement in detail preservation compared to the baseline VN. Quantitatively, it manages to bridge the reconstruction gap reported in VN with the non-equivariant baseline, occNet. Qualitatively, it can be seen the the proposed method is able to preserve some details such as the vehicle side mirror.
- The paper compares performance with the baseline VN on many additional applications including compression, normal estimation, classification, segmentation and registration. In all of them if demonstrates superior performance.

**Weaknesses:**

- The paper's presentation is one of the main reasons for my low rating. It is possible that I did not understand some important insights, but given my relatively broad background in the field, this suggests that the authors did not do an adequate job of presenting and conveying their ideas to the reader. Key issues include the lack of a clear motivation, insufficient highlighting of limitations in existing solutions, inadequate consideration of the suggested design choices, and—most importantly—an outdated comparison with relevant literature
- The description of the limitation of VN is not clear. VN embeds features in a high dimensional space. But instead of using a n-dim vector, they use a nx3 matrix to keep the rotation operation in its same form as in the input space. It is thus my understanding that the representation capacity shouldn’t be smaller than any other global-embedding network which maps inputs to a high n-dim space. Therefore, i couldn’t quite understand the authors claim about VN having “3dim” features. First, I don’t think this is an accurate description of the dimensionality of VN, but more importantly i don’t understand how the fact that the vectors in the list live in 3 dimensions is causing issues with representing geometric details. I’ll try to give an intuitive example. Imagine a very dense pointcloud that captures fine details of the shape. Such a pointcloud may have 10^6 points and these points may live in 3D. My point is, the fact that the points live in R^3 isn’t itself a problem for capturing these details. Instead, what is known to cause issues with capturing fine details in neural fields is global representations. To fight this, many methods focus on partitioning the shape or scene into smaller regions and representing each of them with a local function, like switching from occnet to conv-occnet. In fact, there have been several follow up works to VN that try to do that. Other works tried to improve the encoder too. These works are not mentioned in the submission but should be discussed and compared with: [1,2,3,4].
- It is perhaps my misunderstanding, but it seems the lifting to R^N replaces the principal axes in R^3 with a 3-dim subspace in R^N. Why then is this helpful?
- The compression experiment is a bit unclear to me. There’s no report of the compression ratio — how much is the original pointcloud compressed wrt the embedding? It seems to measure reconstruction on the train set rather than compression.

**Minor**:

- The presentation should be more accurate. Sentences like “Equivariant neural networks (NN) change the output accordingly when the point cloud input is rotated without additional training.” are not accurate.  Equivariant NN are more general than that. Here the authors refer specifically to point cloud networks that are rotation equivariant.
- why is z/z not shown for part segmentation?

Typos:
* theoremIf

**References**:

[1] VN-Transformer: Rotation-Equivariant Attention for Vector Neurons

[2] VNT-Net: Rotational Invariant Vector Neuron Transformers

[3] 3D Equivariant Graph Implicit Functions

[4] On the universality of rotation equivariant point cloud networks

**Questions:**

- which experiment is meant to demonstrate that the lack of details is due to the 3 dim feature vector? Take the non-equivariant network occNet. This network maintains better details and still has N dim. I therefore am not convinced that the issue is with the dim of the features.
- I understanding that the motivation in the proposed lifting to k-dim is to keep it simpler than other tensor networks, but how is the proposed representation compare to it? is it less expressive? i would be glad to see a discussion

---

> ### Author Response · Authors · 2023-11-20
>
> **Weaknesses:**
>
> > **Weakness 1.** The paper's presentation is one of the main reasons for my low rating. It is possible that I did not understand some important insights, but given my relatively broad background in the field, this suggests that the authors did not do an adequate job of presenting and conveying their ideas to the reader. Key issues include the lack of a clear motivation, insufficient highlighting of limitations in existing solutions, inadequate consideration of the suggested design choices, and—most importantly—an outdated comparison with relevant literature
> >
>
> ---
>
> We apologize for the confusion. Let us explain our motivation and limitations of existing methods:
>
> - Previous SO(3) equivariant networks either use
>     - (1) an expressive high-dimensional frequency-based feature representation but a specific architecture to guarantee equivariance (TFN, SE(3)-transformers). This makes it difficult to utilize the state-of-the-art architectures.
>     - (2) a flexible architecture (Vector neurons) but must be used with low-dimensional feature representation that cannot capture details in 3D shapes (explained in depth in subsequent questions).
> - It is possible to use the feature representation used in TFN or SE(3)-transformers together with Vector Neurons. Another alternative is to use Fourier basis features, which is another frequency-based feature representation, with Vector Neurons.
> - However, we found it is difficult to take certain components from methods referenced in (1) without a background in representation theory and quantum mechanics (e.g. the use of Wigner-D matrix and Clebsch-Gordan (CG) product in particular). Fourier features, on the other hand, are intuitive but do not guarantee equivariance.
> - Our contribution is an intuitive and expressive frequency-based feature representation that is provably equivariant and can be used with a flexible architecture, namely Vector Neurons.
>
> For the outdated comparison, we have conducted additional comparison experiments which is described in the response to Weakness 2-2.
>
> > **Weakness 2-1.** The description of the limitation of VN is not clear. VN embeds features in a high dimensional space. But instead of using a n-dim vector, they use a nx3 matrix to keep the rotation operation in its same form as in the input space. It is thus my understanding that the representation capacity shouldn’t be smaller than any other global-embedding network which maps inputs to a high n-dim space. Therefore, i couldn’t quite understand the authors claim about VN having “3dim” features. First, I don’t think this is an accurate description of the dimensionality of VN, but more importantly i don’t understand how the fact that the vectors in the list live in 3 dimensions is causing issues with representing geometric details. I’ll try to give an intuitive example. Imagine a very dense pointcloud that captures fine details of the shape. Such a pointcloud may have 10^6 points and these points may live in 3D. My point is, the fact that the points live in R^3 isn’t itself a problem for capturing these details.
> >
>
> ---
>
> What we meant by 3-dimensional feature is that the feature representation of ***each point in the point cloud*** has 3 dimensions. Indeed, because you have $n$ number of points, the total input may have size $n*3$. However, the fact that the feature of each point is limited to 3 is a problem, as it limits the expressivity especially when we are describing 3D shapes. This is a well-known issue and is referred to as *spectral bias,* which states that neural networks tend to favor low-frequency functions, making the approximation of high-frequency details challenging [1]. One solution to this problem is to map the 3D input to a higher dimensional space using a Fourier basis as done in several papers, including NeRF [2]. However, this mapping does not guarantee equivariance. Our approach can be seen as feature mapping that enables capturing multiple frequencies, with an equivariance guarantee. We have a more in-depth explanation below on why 3-dimensional input is a problem. Please read on if you think this rather short explanation is unsatisfying.

---

> ### Author Response · Authors · 2023-11-20
>
> Consider a simple MLP-based neural network. The capacity to express high-frequency functions (i.e. functions whose output changes a lot even when there is a small change in the input) is limited by the norm of the linear weights. To illustrate this, for two data points $(x_1, y_1)$ and $(x_2, y_2)$ in $(\mathbb{R}^d, \mathbb{R})$, the MLP-based network with ReLU activation can be expressed in piecewise linear relation  $y_1 = A \cdot x_1+b$ and $y_2 = A \cdot x_2+b$ in the active region determined by ReLU, where $A$ and $b$ are weights of network. High-frequency function requires a rapid change in output relative to the input, necessitating large gradient magnitudes. For instance, if $x_1$ and $x_2$ are close yet $y_1$ and $y_2$ differ significantly, then the network should satisfy $|y_2-y_1| = |A \cdot (x_2-x_1)| < |A| \cdot |x_2-x_1|$. The gradient's magnitude is thus constrained by the weight norm due to this linear relationship. Typically, weights are initialized with a certain norm value, hindering the network's ability to approximate high-frequency functions. To effectively represent these functions, the weights must increase their norm, achievable through gradient descent. This accounts for the slower learning of high-frequency functions. This issue is called a spectral bias, and more detailed derivation is available in [1]. This spectral bias explains why approximating high-frequency functions with a neural network is difficult.
>
> To address this issue, one strategy is to preprocess the input data with a function that contains a variety of frequencies. For example, Fourier features map low-dimensional input data to a high-dimensional space using a multi-frequency sinusoidal function. To see why this is effective, consider the following example.
>
> 1. We aim to approximate a function $f$ using an MLP, trained on a dataset of data points $\{(x_i, y_i)\}_{i=1}^N$, where $N$ is the number of data points.
> 2. Consider two closely positioned data points with a steep difference in their outputs, such as $(x_1, y_1) = (0, 0)$ and $(x_2, y_2) = (0.001, 1)$. The slope, or the magnitude of the gradient between these points, is 1000, indicating a high-frequency component in the target function.
> 3. An MLP may struggle to minimize the error for these points due to spectral bias, which makes it inherently challenging for the network to learn high-frequency functions.
> 4. By introducing Fourier features, such as $\psi(x) = [\sin(x), \sin(500\pi x)]$, we transform the input space.
> 5. Under this transformation, the data points are mapped to $([0, 0], 0)$ and $(\psi(0.001), 1) \approx ([0.001, 1], 1)$.
> 6. Consequently, the effective slope in the transformed feature space is reduced to 1, a significant decrease from the original slope of 1000.
> 7. This transformation by the Fourier feature '*spreads out*' the steep variations in the low-dimensional input space, transforming points that would produce a high-frequency response into smoother, more gradual changes.
> 8. As a result, the MLP can now more easily minimize errors at these points, which aligns with the concept of spectral bias—networks are better at learning smoother functions with lower-frequency components.
>
> Our feature representation, FER, shares this same intuition but also guarantees equivariance while Fourier features cannot.
>
> [1] Rahaman, Nasim, et al. "On the spectral bias of neural networks." *International Conference on Machine Learning*. PMLR, 2019.
>
> [2] Mildenhall, Ben, et al. "Nerf: Representing scenes as neural radiance fields for view synthesis." *Communications of the ACM* 65.1 (2021): 99-106.
>
> [3] Tancik, Matthew, et al. "Fourier features let networks learn high frequency functions in low dimensional domains." *Advances in Neural Information Processing Systems* 33 (2020): 7537-7547.

---

> ### Author Response · Authors · 2023-11-20
>
> > **Weakness 2-2.** Instead, what is known to cause issues with capturing fine details in neural fields is global representations. To fight this, many methods focus on partitioning the shape or scene into smaller regions and representing each of them with a local function, like switching from occnet to conv-occnet. In fact, there have been several follow up works to VN that try to do that. Other works tried to improve the encoder too. These works are not mentioned in the submission but should be discussed and compared with: [1,2,3,4].
> >
>
> ---
>
> Our feature representation, FER, can be integrated into any one of the methods you cited. We applied our method to one of the recent VN-based methods you referenced, GraphONet [1], and observed enhanced performance, reinforcing the effectiveness of our approach. The result for GraphONet is 56.0% and 57.6% for FER-GraphONet, which shows 1.6% improvement. The results demonstrated that our proposed feature representation further improves performance when used with the state-of-the-art methods you cited. The code for this experiment is also publicly available, and we will include this result in the paper.
>
> Details of this experiment:
>
> We select 5 meshes from Stanford 3D scanning repository [2], and generate a dataset with 300 surface points and 1024 query points to evaluate occupancy. All meshes are normalized in scale, and we use query points generated by adding small Gaussian noise to the surface points. We train GraphONet baseline and FER-GraphONet, which is a variant of GraphONet by integrating our proposed feature representation (FER). We apply a feature dimension of 8 (3+5). To evaluate representation power, we have adopted a test setting of the occupancy network, where reconstruction quality is evaluated on the meshes used in training. For the evaluation metric, we use volumetric intersection over union (IoU).
>
> [1] Chen, Yunlu, et al. "3d equivariant graph implicit functions." *European Conference on Computer Vision*. Cham: Springer Nature Switzerland, 2022.
>
> [2] The Stanford 3D Scanning Repository, https://graphics.stanford.edu/data/3Dscanrep/
>
> > **Weakness 3.** It is perhaps my misunderstanding, but it seems the lifting to R^N replaces the principal axes in R^3 with a 3-dim subspace in R^N. Why then is this helpful?
> >
>
> ---
>
> In our response to Weakness 2-1, we explained why multi-frequency features such as ours are important. In this context, our method's approach of “lifting to R^N replaces the principal axes in R^3 with a 3-dim subspace in R^N” is not just mapping vectors to higher-dimensional subspace, but doing it in a way that captures multiple frequencies, as evident in Theorem 2, all the while guarantee equivariance.
>
> > **Weakness 4.** The compression experiment is a bit unclear to me. There’s no report of the compression ratio — how much is the original pointcloud compressed wrt the embedding? It seems to measure reconstruction on the train set rather than compression.
> >
>
> ---
>
> Thank you for pointing this out. We acknowledge the confusion stemming from the use of the term "compression" and appreciate the opportunity for clarification. In our context, we were primarily interested in measuring the representation capability of different networks, which can be measured using the reconstruction performance.
>
> To address your concern regarding compression ratio, we have conducted an additional experiment where we match the embedding size of all the methods, thus having the same compression ratio, and try to evaluate their reconstruction (i.e. decompression) performance. In the result, we observe that our method achieves the highest IoU while having the same compression ratio.
>
> | Model type  | Input size | Embedding size | Compression ratio | IoU (%) |
> | --- | --- | --- | --- | --- |
> | OccNet | 300 x 3 = 900 | 513 | 57% | 67.5 |
> | VN-OccNet  | 300 x 3 = 900 | 171 x 3 = 513 | 57% | 73.4 |
> | FER-OccNet (n=3+5) | 300 x 3 = 900 | 64 x 8 = 512 | 57% | 77.3 |
>
> > **Minor weakness 1.** The presentation should be more accurate. Sentences like “Equivariant neural networks (NN) change the output accordingly when the point cloud input is rotated without additional training.” are not accurate. Equivariant NN are more general than that. Here the authors refer specifically to point cloud networks that are rotation equivariant.
> >
>
> ---
>
> You are correct. Not all equivariance is about rotations. We will correct these.
>
> > **Minor weakness 2.** why is z/z not shown for part segmentation?
> >
>
> ---
>
> Because the purpose is to demonstrate SO(3)-equivariant capability of different models, which can only be assessed in z/SO(3) and SO(3)/SO(3).

---

> ### Author Response · Authors · 2023-11-20
>
> **Questions:**
>
> > **Question 1.** Which experiment is meant to demonstrate that the lack of details is due to the 3 dim feature vector? Take the non-equivariant network occNet. This network maintains better details and still has N dim. I therefore am not convinced that the issue is with the dim of the features.
> >
>
> ---
>
> The limited capacity of low dimensional features has been well-studied in other papers (See our response to Weakness 2-2, and Figure 1 in [1]), and we figured this would be out of the scope of our paper.
>
> The fact that VN-OccNet is outperformed by non-equivariant OccNet is already noted in the original Vector Neuron paper, and this is exactly because of their limited feature representation. Our paper is a work towards fixing this.
>
> [1] Tancik, Matthew, et al. "Fourier features let networks learn high frequency functions in low dimensional domains." *Advances in Neural Information Processing Systems* 33 (2020): 7537-7547.
>
> > **Question 2.** I understanding that the motivation in the proposed lifting to k-dim is to keep it simpler than other tensor networks, but how is the proposed representation compare to it? is it less expressive? i would be glad to see a discussion.
> >
>
> ---
>
> We actually learn a better representation and achieve better results. For instance, our classification and segmentation results indicate that our networks perform better than Tensor Field Networks (TFN). This is because we can flexibly choose recent advances in point processing networks such as DGCNN [1], while TFN must use a specific architecture to preserve equivariance.
>
> [1] Wang, Yue, et al. "Dynamic graph cnn for learning on point clouds." *ACM Transactions on Graphics (tog)* 38.5 (2019): 1-12.

---

> ### Comment · Reviewer_Jvrz · 2023-11-22
> **Equivariance of Fourier Features**
>
> When the authors say Fourier features are not equivariant, what do they mean? The representations of a function in the harmonics of $S^2$ or $SO(3)$ are equivariant to the $SO(3)$ group action since $S^2$ and $SO(3)$ are homogeneous spaces of $SO(3)$. Perhaps it is meant that representations of a function in the harmonics of $S^1$ are not equivariant to the $SO(3)$ group action? Maybe I'm misunderstanding.

---

> > ### Author Response · Authors · 2023-11-23
> >
> > Thank you for your insightful query, and we apologize for any confusion caused by our initial statement. In our discussion about Fourier features not being equivariant, we specifically refer to scalar Fourier features. One example of this is $enc(x)=(\sin(2^0\pi x),...,\sin(2^{L-1}\pi x),\cos(2^0\pi x),...,\cos(2^{L-1}\pi x) )$, where $x \in \mathbb{R}$ and $L \in \mathbb{N}$, as detailed in references [1], [2]. We can apply these features to each coordinate of a point input. This form of feature representation, focusing on individual scalar inputs, is not equivariant under the $SO(3)$ group action, yet it has shown effectiveness in enhancing the ability to capture high-frequency details in shapes.
> >
> > [1] Mildenhall, Ben, et al. "Nerf: Representing scenes as neural radiance fields for view synthesis." *Communications of the ACM* 65.1 (2021): 99-106.
> >
> > [2] Tancik, Matthew, et al. "Fourier features let networks learn high frequency functions in low dimensional domains." *Advances in Neural Information Processing Systems* 33 (2020): 7537-7547.

---

### Official Review · Reviewer_XaCw · 2023-10-30

**Soundness:** 3 good
**Presentation:** 3 good
**Contribution:** 3 good
**Rating:** 6
**Confidence:** 4

**Summary:**

The paper introduces an equivariant feature representation to map per 3D point to a high-dimensional feature vector, which is supposed to be more capable to represent rich point information. Extensive experiments are conducted to verify the effectiveness in multiple tasks: point cloud completion, 3D shape compression, normal estimation, point cloud registration, and point cloud classification and segmentation. The results are very positive.

**Strengths:**

The primary strength of the proposed component lies in its capability to map 3D points to high-dimensional feature space with the property of SO(3)-equivariant, while the existing work Vector Neuron(VN) can only deal with three-dimensional features. The construction of the frequency-based transformation function D in equation (1) looks very reasonable and provable.

Secondly, the effectiveness of the proposed module has been extensively evaluated on multiple tasks and datasets, which makes this work very solid and convincing.

**Weaknesses:**

There are some minor questions.

1. The proposed component is always applied together with the existing VN module. Is it possible to work by its alone? If so, how to extend it?

2. Since the construction of transformation function D is related to different frequencies. However, in the evaluation, there is a lack of concrete experiments to deeply analyse how such frequency-based function can help the network to learn more discriminative features. Although most of downstream tasks have excellent performance, it is unclear what type of features are learned while keeping the SO(3) equivariance.

3. All experiments are conducted on very small-scale point clouds like objects. Is it possible to scale up the proposed method on larger scale 3D point clouds such as room-level ScanNet/S3DIS datasets or even urban-level SensatUrban dataset? If not, what are the potential reasons?

4. In Section 2.2, the last line states that "Our feature representation can xxxx, which is not equivariant, so that it is equivariant". It's suggested to clarify the point?

5. In page 4, the section "Theorem 1.", you may need to remove the word "theorem"?

**Questions:**

Details in weaknesses.

---

> ### Author Response · Authors · 2023-11-20
>
> **Weaknesses:**
>
> > **Weakness 1.** The proposed component is always applied together with the existing VN module. Is it possible to work by its alone? If so, how to extend it?
> >
>
> ---
>
> Yes, the proposed component can operate without VN module. For instance, in our spherical shape regression experiment (see Appendix G.2), we demonstrated this capability without relying on the VN module.
>
> > **Weakness 2.** Since the construction of transformation function D is related to different frequencies. However, in the evaluation, there is a lack of concrete experiments to deeply analyse how such frequency-based function can help the network to learn more discriminative features. Although most of downstream tasks have excellent performance, it is unclear what type of features are learned while keeping the SO(3) equivariance.
> >
>
> ---
>
> Thank you for pointing this out. I think this is an important analysis that we have missed. To do this, we have an additional experiment on shape completion that compares which features are responsible for completing which part of the shape. We would expect that the low-frequency features would pick up on low-frequency regions of the shape (i.e. regions where shape does not change much), and vice-versa.
>
> Our composite feature is a concatenation of $n=3,5,7,9$ dimensional features (See Appendix E.1.1 for how we make the concatenation). The composite feature is then processed through a three-layer MLP to predict occupancy values in an occupancy network. To visualize the contribution of each feature dimension on the completed shape, we compute the gradient magnitude of the occupancy predictions with respect to our feature representation of the surface points on the reconstructed mesh for each $n$. More concretely, we have $\psi_3(x), \psi_5(x), \psi_7(x), \psi_9(x)$ where $x \in R^3$ is a surface point and $\psi_n(x)$ ($R^3→R^n$) is our feature presentation with $n$ number of dimensions. These four features are concatenated and processed by MLP to predict occupancy $o=f(\phi_3,\phi_5,\phi_7,\phi_9)$ where $f$ is MLP. Then, gradient used for visualization is $\partial o/ \partial \phi_3$, $\partial o / \partial \psi_5$, $\partial o / \partial \psi_7$, $\partial o / \partial \psi_9$.
>
> This visualization was performed individually for each feature. The greater the magnitude of the gradient, the more yellow it is.
>
> [Anonymous link for the visualization 1](https://drive.google.com/file/d/16AMhfh-776A3xWXtOntDNGE73i7gBhUu/view?usp=sharing)
>
> [Anonymous link for the visualization 2](https://drive.google.com/file/d/1sGpR5L6kGF0DvLsdGDJCJyfe4TdlUvGb/view?usp=sharing)
>
> [Anonymous link for the visualization 3](https://drive.google.com/file/d/1NzoRLiLpYIshvFQwZKD-EY2ExDIulm71/view?usp=sharing)
>
> Our findings indicate that the lower-dimensional features, particularly for *n*=3 and *n*=5, predominantly capture the more expansive and volumetric components of the meshes, such as the torso and back of these objects. In contrast, the higher-dimensional features at *n*=7 and *n*=9 tend to focus on finer and more intricate details, such as facial features. These results offer valuable insights into the network's ability to differentiate between broad structural elements and detailed features of the meshes, reinforcing the effectiveness of frequency-based feature mappings in an SO(3) equivariant learning context.
>
> [1] The Stanford 3D Scanning Repository, https://graphics.stanford.edu/data/3Dscanrep/
>
> > **Weakness 3.** All experiments are conducted on very small-scale point clouds like objects. Is it possible to scale up the proposed method on larger scale 3D point clouds such as room-level ScanNet/S3DIS datasets or even urban-level SensatUrban dataset? If not, what are the potential reasons?
> >
>
> ---
>
> While our method is applicable to scene-level tasks, we focused on object orientations because rotating the entire scene is less common in practice — you typically have oriented objects, not scenes. This is because in most cases the floor orientation defines the canonical orientation for scenes, and you can simply detect the floor and correct it to the canonical orientation.
>
> > **Weakness 4.** In Section 2.2, the last line states that "Our feature representation can xxxx, which is not equivariant, so that it is equivariant". It's suggested to clarify the point?
> >
>
> ---
>
> We have clarified it as “However, conventional Fourier basis is not SO(3) equivariant. Our feature representation guarantees equivariance, **and** captures multiple frequencies“

---

> > ### Comment · Reviewer_XaCw · 2023-12-03
> > **Thanks**
> >
> > Thanks for the authors' feedback and I will keep my positive score.

---

### Official Review · Reviewer_79hQ · 2023-10-30

**Soundness:** 3 good
**Presentation:** 3 good
**Contribution:** 3 good
**Rating:** 6
**Confidence:** 3

**Summary:**

The paper targets the challenging 3D representation problem and a new equivariant feature representation which can map 3D points to high-dimensional feature space is presented. The experimental results have been validated on several tasks like point-cloud completion, shape compression, normal estimation, point cloud registration, point cloud classification and segmentation. Reasonable results have been reported with a comparision with the existing work.

**Strengths:**

1. A new 3D representation is presented which can discern multiple frequencies in the 3D data. Also, it provides theretical justification of the proposed approach.
2. The proposed approach is valiated on several tasks like point-cloud completion, shape compression, normal estimation, point cloud registration, point cloud classification and segmentation. Competitive performances have been reported with a comparison of the existing baselines.

**Weaknesses:**

1. For the experiments on point-cloud completion, as the experimental setup discussed, it samples 300 points for the evaluation. How about the performance of different number of sampled points?
2. For the experiments on the test classification and part segmentation, it seems the proposed algorithm does not have obvious performance gain over the existing work like PaRINet. What is the main reason under this result?
3. A minor suggestion, the baselines are usually introduced in the year before 2022. Is it possible to report more recent results for the comparison in the experimental section, like the work published in 2023?

**Questions:**

Please address the questions in the weakness section.

---

> ### Author Response · Authors · 2023-11-20
>
> **Weaknesses:**
>
> > **Weakness 1.** For the experiments on point-cloud completion, as the experimental setup discussed, it samples 300 points for the evaluation. How about the performance of different number of sampled points?
> >
>
> ---
>
> We have conducted further experiments to assess the impact of the number of points. We train models with a dataset where each data consists of 300 points. We then evaluated the models using 200 to 600 input points during test time. As illustrated in the following figure, our method is more robust to changes in the number of points compared to both the Vector Neuron (VN)-based and vanilla Occupancy networks.
>
> [anonymous link for the figure](https://drive.google.com/file/d/16o-XEmPTiuBkeEd00kD8glcxZ3wQCuyp/view?usp=sharing)
>
> The reason why performance goes down as the number of points increases beyond ~350 input points is because these methods extract point-wise features from a point cloud. VN uses K-Nearest-Neighbor(KNN) for each point-wise feature initialization, which processes the relative coordinates of adjacent points. Relative coordinates are sensitive to changes in point density. Such changes alter the scale of relative coordinates compared to what was observed during training. This change will decrease the performance not only when the number of points gets smaller but also larger.
>
> > **Weakness 2.** For the experiments on the test classification and part segmentation, it seems the proposed algorithm does not have obvious performance gain over the existing work like PaRINet. What is the main reason under this result?
> >
>
> ---
>
> Yes for those tasks, PaRINet showed the best performance and ours came second. This is because PaRINet is a rotation-***invariant*** model tailored for tasks such as part segmentation. However, it is less general than our method because it cannot be applied to rotation-equivariant tasks. Equivariance is more general because you can create invariance from equivariance, but not in the other direction. Our model shows the best performance among the rotation equivariant models.
>
> > **Weakness 3.** A minor suggestion, the baselines are usually introduced in the year before 2022. Is it possible to report more recent results for the comparison in the experimental section, like the work published in 2023?
> >
>
> ---
>
> We have reviewed SO(3) equivariant networks published in 2023 such as [1][2], but we found that they perform worse than Vector Neurons, and figured that it would not be necessary. However, we would be happy to hear your thoughts on this.
>
> [1] S´ekou-Oumar Kaba, Arnab Kumar Mondal, Yan Zhang, Yoshua Bengio, and Siamak Ravanbakhsh.
> Equivariance with learned canonicalization functions. In International Conference on Machine
> Learning, pp. 15546–15566. PMLR, 2023.
>
> [2] Zhu, Minghan, et al. "E2PN: Efficient SE (3)-equivariant point network." *Proceedings of the IEEE/CVF Conference on Computer Vision and Pattern Recognition*. 2023.

---

> > ### Comment · Reviewer_79hQ · 2023-11-22
> >
> > I would appreciate the replies from the authors. My concerns in the last round have been addressed. I will maintain the positive view on the rating.

---

### Official Review · Reviewer_MkmG · 2023-10-31

**Soundness:** 3 good
**Presentation:** 3 good
**Contribution:** 3 good
**Rating:** 6
**Confidence:** 3

**Summary:**

The paper introduces a frequency-based input representation for Vector-Neuron for SO-3 equivariant network. It provides compelling evidence to support the claim of keeping equivariance with this input representation. Both visualized and numerical results demonstrate the effectiveness of the proposed method, thus validating its ability to achieve equivariant representations.

**Strengths:**

- The paper presents sufficient mathematical proofs, which provide robust support for the proposed representation. This strong evidence reinforces the credibility of the approach.
- The multi-frequency design demonstrates its effectiveness in terms of both visualization and benchmark performance.
- The writing style and figures in the paper are great as they effectively elucidate the design and motivation behind the proposed representation.
- The experiments conducted in the study convincingly demonstrate the effectiveness of the proposed method across various downstream tasks.

**Weaknesses:**

- Lack of ablation studies
- (This is not a weakness) Does author try to extend the method on scene (indoor) ?

**Questions:**

See above

---

> ### Author Response · Authors · 2023-11-20
>
> **Weaknesses:**
>
> > **Weakness 1.** Lack of ablation studies
> >
>
> ---
>
> Indeed our initial submission did not sufficiently address the impact of the dimensionality of our feature representation on the computational cost and performance, so we conducted additional experiments in the shape compression task. Below are the results for processing 300 points for a single object for encoding and 100,000 query points for decoding. The results are averaged over 300 predictions. All experiments used A6000 GPU.
>
> | Model type | IoU (%) | inference time - encoder | inference time - decoder |
> | --- | --- | --- | --- |
> | VN-OccNet | 73.4 | 3.44 $\pm$ 0.05 ms | 9.57 $\pm$ 0.10 ms |
> | FER-VN-OccNet (n=8) | 81.0 | 3.47 $\pm$ 0.06 ms | 9.64 $\pm$ 0.16 ms |
> | FER-VN-OccNet (n=15) | 81.9 | 6.55 $\pm$ 0.26 ms | 9.87 $\pm$ 0.21 ms |
>
> As the table shows,  incorporating our 8-dimensional feature representation (FER-VN-OccNet (n=8)) boosts performance (a 7.6% increase in IoU) with a negligible impact on encoder inference time compared to the VN-OccNet baseline. However, expanding the feature representation to 15 dimensions (FER-VN-OCCNet (n=15)) doubles the computational time of the encoder with only a marginal performance improvement. On the other hand, the decoder inference time remained relatively stable across models, underscoring the efficiency of integrating FER into the network.
>
> **Comments:**
>
> > **Comment 1.** (This is not a weakness) Does author try to extend the method on scene (indoor) ?
> >
>
> ---
>
> While our method is applicable to scene-level tasks, we focused on object orientations because rotating the entire scene is less common in practice — you typically have oriented objects, not scenes. This is because in most cases the floor orientation defines the canonical orientation for scenes, and you can simply detect the floor and correct it to the canonical orientation.

---

> > ### Comment · Reviewer_MkmG · 2023-12-03
> > **Keep my score**
> >
> > I appreciate the author's thorough response to my questions. After revisiting the paper and the rebuttal, I have decided to maintain my current rating.

---

### Official Review · Reviewer_Jvrz · 2023-11-05

**Soundness:** 2 fair
**Presentation:** 2 fair
**Contribution:** 3 good
**Rating:** 5
**Confidence:** 2

**Summary:**

In this paper the authors propose a representation for point cloud data that can be used with the existing method [1] to improve expressivity while maintinaing equivariance to 3D rotations. The representation is constructed by transforming points to an axis-angle representation, then lifting that representation to a random subspace of SO(n). The subspace is defined by three matrices J_1, J_2, and J_3, where J_1 is a random vector in the Lie algebra of SO(n), and J_2 and J_3 (also in the Lie algebra) are matrices whose Lie bracket is close to J_1. The final SO(n) representation of a given point is the exponential map of the linear combination of J_1, J_2, and J_3, where the combination coefficients are determined by the axis-angle representation.

[1] Deng, Congyue, et al. "Vector neurons: A general framework for so (3)-equivariant networks." Proceedings of the IEEE/CVF International Conference on Computer Vision. 2021.

**Strengths:**

*Originality:* The proposed approach appears novel. Although other approaches have used a Fourier like representation to represent 3D objects [1,2,3], this approach differs from those in that 1) the representation is intentionally high-dimensional, 2) only a subspace of the high-dimensional space is considered.

*Quality:* In the experiments, proposed method performs on-par with or better than most baselines.

*Clarity:* (see weakness)

*Significance:* Designing expressive equivariant representations is a challenging and important problem in the field.


[1] Esteves, Carlos, et al. "Learning so (3) equivariant representations with spherical cnns." Proceedings of the European Conference on Computer Vision (ECCV). 2018.

[2] Cohen, Taco S., et al. "Spherical cnns." arXiv preprint arXiv:1801.10130 (2018).

[3] Kondor, Risi, Zhen Lin, and Shubhendu Trivedi. "Clebsch–gordan nets: a fully fourier space spherical convolutional neural network." Advances in Neural Information Processing Systems 31 (2018).

**Weaknesses:**

*Quality:*
* The authors present a very critical view of methods that use spherical harmonics which I think is not only inappropriate but also incorrect. Spherical harmonics can be seen analogous to the Fourier basis which have been applied in signal processing problems for centuries, and the authors rely on for intuition building. Also, the authors state that spherical harmonics come from quantum mechanics, but the use of spherical harmonics predates the "discovery" of quantum mechanics by more than a century.
* There does not appear to be any discussion of the impact of choosing a higher dimensional input representation space. How does the choice of $n$ impact performance? What is the additional computational cost?
* The notation is inconsistent (see questions)
* Some proofs are missing (see questions)

*Clarity:* I found the presentation to be difficult to follow. The paper uses advanced math concepts such as the Lie bracket without motivating or even naming them.

**Questions:**

*Questions:*
- What is meant by “$R^z(\hat{u})$ is a rotation matrix defining the orientation measured from the $z$-axis to $\hat{u}$.” Is it the 2D rotation about the axis $z \times \hat{u}$ that aligns the $z$-axis and $\hat{u}$?
- It seems like the map is from SO(3) to a subspace of SO(n) is that right
- Is the axis arbitrarily selected?
- Createsearchspace from Algo 2 is not described in the text, how does this work?

*Possible typos:*
- “Theorem 1. theoremIf” → “Theorem 1. If”
- Are $\hat{u}$ and $\overrightarrow{u}$ used interchangeably in section 3? It would be clearer if the notation were consistent
- “intuitively, just like F1, F2 and F3 represent angles” → “intuitively, just like F1, F2 and F3 represent axes of rotation”
- Should “axes ψ(ˆx), ψ(ˆy), and ψ(ˆz)” be “axes ψ(F_1), ψ(F_2), and ψ(F_3)”? If not what does $\hat{.}$ mean here?
- “ effective-yet-rotation-equivariant” →  expressive yet rotation equivariant
- In section D, readers are referred to the proof of Thm 1 for the proof of Prop 1, but the proof of Thm 1 does not prove Prop 1.

---

> ### Author Response · Authors · 2023-11-20
>
> **Weaknesses:**
>
> > **Weakness 1.** The authors present a very critical view of methods that use spherical harmonics which I think is not only inappropriate but also incorrect. Spherical harmonics can be seen as analogous to the Fourier basis which has been applied in signal processing problems for centuries, and the authors rely on intuition building. Also, the authors state that spherical harmonics come from quantum mechanics, but the use of spherical harmonics predates the "discovery" of quantum mechanics by more than a century.
> >
>
> ---
>
> Our sincere apologies for the confusion caused by our previous statement. To clarify, we did not intend to say spherical harmonics were developed for quantum mechanics. Our reference was specifically to the Wigner-D matrix and Clebsch-Gordan (CG) product, which, as far as our knowledge goes, are used to describe quantum states.
>
> In one of the very first papers on SO(3) equivariant neural networks [1], the authors give a brief description of the Wigner-D matrix and its specific operation (CG product) and delegate the in-depth explanation to [2], a quantum mechanics textbook. They also rely on representation theory for explaining equivariant mapping in relation to the Wigner-D matrix and CG product. While these pieces of knowledge may have been self-evident for the authors of [1], whose background is in physics, we personally found them difficult to grasp and wished for a more intuitive approach. And we figured that machine learning and computer vision communities would share the same sentiment, and benefit from a more intuitive approach which motivated this work.
>
> We will definitely tone down our statements. We do have great respect for these earlier works, and our language just came out more aggressive than intended.
>
> [1] Thomas, Nathaniel, et al. "Tensor field networks: Rotation-and translation-equivariant neural networks for 3d point clouds." *arXiv preprint arXiv:1802.08219* (2018).
>
> [2] Bransden, Brian Harold, and Charles Jean Joachain. "Introduction to quantum mechanics." (1989).
>
> > **Weakness 2.** There does not appear to be any discussion of the impact of choosing a higher dimensional input representation space. How does the choice of $n$ impact performance? What is the additional computational cost?
> >
>
> ---
>
> Indeed our initial submission did not sufficiently address the impact of the dimensionality of our feature representation on the computational cost and performance, so we conducted additional experiments in the shape compression task. Below are the results for processing 300 points for a single object for encoding and 100,000 query points for decoding. The results are averaged over 300 predictions. All experiments used A6000 GPU.
>
> | Model type | IoU (%) | inference time - encoder | inference time - decoder |
> | --- | --- | --- | --- |
> | VN-OccNet | 73.4 | 3.44 $\pm$ 0.05 ms | 9.57 $\pm$ 0.10 ms |
> | FER-VN-OccNet (n=8) | 81.0 | 3.47 $\pm$ 0.06 ms | 9.64 $\pm$ 0.16 ms |
> | FER-VN-OccNet (n=15) | 81.9 | 6.55 $\pm$ 0.26 ms | 9.87 $\pm$ 0.21 ms |
>
> As the table shows,  incorporating our 8-dimensional feature representation (FER-VN-OccNet (n=8)) boosts performance (a 7.6% increase in IoU) with a negligible impact on encoder inference time compared to the VN-OccNet baseline. However, expanding the feature representation to 15 dimensions (FER-VN-OCCNet (n=15)) doubles the computational time of the encoder with only a marginal performance improvement. On the other hand, the decoder inference time remained relatively stable across models, underscoring the efficiency of integrating FER into the network.

---

> ### Author Response · Authors · 2023-11-20
>
> **Questions:**
>
> > **Question 1.** What is meant by “$R^z(\hat{u})$ is a rotation matrix defining the orientation measured from the $z$-axis to $\hat{u}$.” Is it the 2D rotation about the axis $z \times \hat{u}$ that aligns the $z$-axis and $\hat{u}$?
> >
>
> ---
>
> Correct. Sorry for the confusion. In the axis-angle representation of $R^z(\hat{u})$, the axis is a unit vector aligned with $\hat{z} \times \hat{u}$  where $\hat{z}=[0,0,1]$ is unit vector for z-axis, and the angle is the one between $\hat{z}$ and $\hat{u}$.
>
> > **Question 2.** It seems like the map is from SO(3) to a subspace of SO(n) is that right?
> >
>
> ---
>
> The map is indeed from the SO(3) group to a subgroup of SO(n).
>
> > **Question 3.** Is the axis arbitrarily selected?
> >
>
> ---
>
> Yes. Axes in high-dimensional space are determined by $J_1, J_2$ and $J_3$, which are arbitrarily sampled from Algorithm 1 $ConstructJ_1,J_2,J_3$.
>
> > **Question 4.** Createsearchspace from Algo 2 is not described in the text, how does this work?
> >
>
> ---
>
> Thank you for pointing out the lack of clarity regarding the "Createsearchspace" function in Algorithm 2. We have described this in Appendix F, but we acknowledge that it was not as clear.
>
> In essence, the "Createsearchspace" function operates as follows:
>
> 1. **Selection of Constraints**: We begin by selecting a subset of constraints from Theorem 1 that can be represented as linear relations concerning $J_1$ and $J_2$. This relation is described in Equation 15 in our paper.
> 2. **Construction of Search Space**: With these linear constraints, we then proceed to construct the search space for $J_1$ and $J_2$. This involves identifying the null space of the linear system formed by these constraints. The null space provides us with a range of possible values for $J_1$ and $J_2$ that satisfy the constraints.
>
> In the revised version of our manuscript, we will elaborate on this process to ensure that the methodology behind the "Createsearchspace" function is transparent and easily understandable.
>
> > **Question 5.** In section D, readers are referred to the proof of Thm 1 for the proof of Prop 1, but the proof of Thm 1 does not prove Prop 1.
> >
>
> ---
>
> We apologize for the confusion. This has been fixed in the updated paper. We separately wrote the proof of Prop 1 in section D.
>
> > **Question 6.** Are $\hat{u}$ and $\vec{u}$ used interchangeably in section 3? It would be clearer if the notation were consistent
> >
>
> ---
>
> They are not interchangeable. $\vec{u} \in \mathbb{R}^3$ is a input point, and $\hat{u} = \vec{u} / ||\vec{u} ||$ is a unit vector of $\vec{u}$, so they are different. However, they both can be applicable for $\psi$. So, for example, given the definition of $\psi(\vec{u})=||\vec{u}||D(R^z(\hat{u}))\hat{e}$,  $\psi(\hat{u})=||\hat{u}||D(R^z(\hat{u}))\hat{e} = D(R^z(\hat{u}))\hat{e}$ since $||\hat{x}||$ is 1.
>
> > **Question 7.** Should “axes ψ(ˆx), ψ(ˆy), and ψ(ˆz)” be “axes ψ(F_1), ψ(F_2), and ψ(F_3)”? If not what does .^ mean here?
> >
>
> ---
>
>  $\hat{x} = [1, 0, 0]$ is a unit vector for the x-axis. And $\psi(\hat{x})= D(R^z(\hat{x}))\hat{e}$ is a unit vector in $\mathbb{R}^n$. Also, $F_i \in \mathbb{R}^{3 \times 3}$ is a matrix and cannot be an input for $\psi(\cdot)$, and as far as we know, we have never used such notation.

---

### Comment · Area_Chair_knbC · 2023-11-21
**Rebuttal Response**

Dear Reviewers,

This is a friendly reminder that the authors have published a rebuttal, and the end of the discussion period is tomorrow.
Due to the tight deadline please take a look at it as soon as possible and check if your questions/comments have been properly addressed.

Thank you again for your time and effort!

AC.

---

### Meta-Review · Area_Chair_knbC · 2023-12-06

**Metareview:**

The paper introduces an equivariant feature representation that maps 3D points to a high-dimensional feature vector, offering increased representation capacity. Extensive experiments using different point-cloud architectures, demonstrate the effectiveness of the features across various point-cloud tasks, including point cloud completion, 3D shape compression, among other. The paper has several theoretical justifications/rationals for the methodology.

Reviewers agree that the method is novel and interesting. Some reviewers expressed concerns about the ablation studies and an inadequate choice of baselines (which didn't include some state of the art methods), which the authors partially addressed resulting in a slight increase of the rating. Similar concerns were raised regarding computational complexity, prompting the authors to provide an additional data point, which is insufficient for conclusive analysis of asymptotic behavior.

The paper's core idea appears novel, and the experiments are comprehensive. However, concerns remain regarding the clarity of some aspects of computational complexity. Additionally, the manuscript contains incorrect statements related to spherical harmonics and quantum mechanics.

**Justification For Why Not Higher Score:**

There are an unresolved issue with the computational complexity (one extra data point is not enough for an asymptotic claim). None of the reviewers provided a rating to merit a higher score.

**Justification For Why Not Lower Score:**

The paper proposes a novel equivariant features, which may be useful of point cloud based ML methods.

---

### Decision · Program_Chairs · 2024-01-16

Accept (poster)